# Does your graph need a confidence boost? Convergent boosted smoothing on graphs with tabular node features

**Jiuhai Chen** *
University of Maryland

**Jonas Mueller & Vassilis N. Ioannidis & Soji Adeshina**
Amazon

**Yangkun Wang**
Shanghai Jiao Tong University

**Tom Goldstein**
University of Maryland

**David Wipf**
Amazon

## Abstract

For supervised learning with tabular data, decision tree ensembles produced via boosting techniques generally dominate real-world applications involving iid training/test sets. However for graph data where the iid assumption is violated due to structured relations between samples, it remains unclear how to best incorporate this structure within existing boosting pipelines. To this end, we propose a generalized framework for iterating boosting with graph propagation steps that share node/sample information across edges connecting related samples. Unlike previous efforts to integrate graph-based models with boosting, our approach is anchored in a principled meta loss function such that provable convergence can be guaranteed under relatively mild assumptions. Across a variety of non-iid graph datasets with tabular node features, our method achieves comparable or superior performance than both tabular and graph neural network models, as well as existing hybrid strategies that combine the two. Beyond producing better predictive performance than recently proposed graph models, our proposed techniques are easy to implement, computationally more efficient, and enjoy stronger theoretical guarantees (which make our results more reproducible).

## 1 Introduction

*Tabular data* consists of observations stored as rows of a table, where multiple numeric/categorical features are recorded for each observation, one per column. Models for tabular data must learn to output accurate predictions solely from (potentially high-dimensional or sparse) sets of heterogeneous feature values. For learning from tabular data, ensembles of decision trees frequently rank on the top of model leaderboards, as they have proven to be highly performant when trained via multi-round boosting algorithms that progressively encourage the learner to focus more on "difficult" examples predicted inaccurately in earlier rounds (Bansal, 2018; Elsayed et al., 2021; Fakoor et al., 2020; Huang et al., 2020b; Ke et al., 2017; Prokhorenkova et al., 2018). Boosted trees thus form the cornerstone of supervised learning when rows of a table are independent and identically distributed (iid).

However, the iid assumption is severely violated for graph structured data (Chami et al., 2020), in which nodes store features/labels that are highly correlated with those of their neighbors. Many methods have been proposed to account for the graph structure during learning, with *graph neural networks* (GNN) becoming immensely popular in recent years (Scarselli et al., 2008; Wang et al., 2019; Zhou et al., 2020a). Despite their success, modern GNNs suffer from various issues (Alon & Yahav, 2021; Oono & Suzuki, 2019): they are complex both in their implementation and the amount of required computation (Faber et al., 2021; Huang et al., 2020a; Shchur et al., 2018), and limited theoretical guarantees exist regarding their performance or even the convergence of their training (Garg et al., 2020; Li & Cheng, 2021; Zhou et al., 2020b). Furthermore, the sample complexity of sophisticated GNN models and their ability to handle rich node features may be worse than simpler models like those used for tabular data (Faber et al., 2021; Huang et al., 2020a; Shchur et al., 2018).

---

* Work done during internship at Amazon Web Services Shanghai AI Lab

Graph datasets store tabular feature vectors, along with additional edge information. Despite the close relationship between graph and tabular data, there has been little work on how to adapt powerful boosting methods – our sharpest tool for dissecting tabular data – to account for edges. Here we consider how to best leverage tabular models for non-iid node classification/regression tasks in graphs. We focus on settings where each node is associated with tabular (numeric/categorical) features $x$ and labels $y$, and the goal is to predict the labels of certain nodes. Straightforward application of tabular modeling (in which nodes are treated as iid examples) will produce a learned mapping where training and inference are uninformed by how nodes are arranged in the graph.

To account for the graph structure, we introduce simple graph propagation operations into the definition of a modified, non-iid boosting loss function, such that edge information can be exploited to improve model accuracy relative to classical alternatives (Friedman, 2001). We refer to the resulting algorithm as EBBS which stands for *efficient bilevel boosted smoothing*, whereby the end-to-end training of a bilevel loss is such that values of the boosted base model $f$ *ebb and flow* across the input graph producing a smoothed predictor $\widetilde{f}$. And unlike an existing adaptation of boosting for graph data (Ivanov & Prokhorenkova, 2021), our approach is simpler (no GNN or other auxiliary model required), produces more accurate predictions on benchmark datasets, and enjoys reliable convergence guarantees.

## 2 BACKGROUND

Consider a graph $\mathcal{G}$ with a set of vertices, i.e. nodes, $\mathcal{V} \triangleq \{v_1, v_2, \ldots, v_n\}$ whose connectivity is described by the edge set $\mathcal{E} \triangleq \{(v_i, v_j) \in \mathcal{V} \times \mathcal{V}\}$, where the tuple of nodes $(v_i, v_j)$ implies a relationship among $v_i$ and $v_j$. The edge set is represented by the adjacency matrix $\boldsymbol{A} = (a_{ij})_{n \times n} \in \mathbb{R}^{n \times n}$, whose entry $a_{ij}$ is nonzero if $(v_i, v_j) \in \mathcal{E}$. In node prediction tasks, each node $v_i$ is associated with a feature vector $\boldsymbol{x}_i \in \mathbb{R}^{d_0}$, as well as a label $\boldsymbol{y}_i \in \mathbb{R}^c$ that might hold discrete or continuous values (corresponding to classification or regression tasks). In certain cases we have access to the labels at only a subset of nodes $\{\boldsymbol{y}_i\}_{i \in \mathcal{L}}$, with $\mathcal{L} \subset \mathcal{V}$. Given $\{\boldsymbol{y}_i\}_{i \in \mathcal{L}}$, the connectivity of the graph $\mathcal{E}$, and the feature values of all nodes $\{\boldsymbol{x}_i\}_{i \in \mathcal{V}}$, our task is to predict the labels of the unlabeled nodes $\{\boldsymbol{y}_i\}_{i \in \mathcal{U}}$, with $\mathcal{U} = \mathcal{V} \setminus \mathcal{L}$. In the most straightforward application, a tabular model may simply be fit to dataset $\{(\boldsymbol{x}_i, \boldsymbol{y}_i)\}_{i \in \mathcal{L}}$, disregarding that the observations are not iid. For prediction, the learned model is then independently applied to each $\boldsymbol{x}_i$ for $i \in \mathcal{U}$. Of course this naive approach may fare poorly as it fails to leverage $\mathcal{E}$, but we note that it empirically performs better than one might expect on certain graph datasets (where perhaps each $\boldsymbol{x}_i$ contains sufficiently rich information to infer the corresponding $\boldsymbol{y}_i$, or nodes connected in $\mathcal{E}$ only exhibit weak correlation in their features or labels) (Faber et al., 2021; Huang et al., 2020a; Ivanov & Prokhorenkova, 2021).

### 2.1 GRADIENT BOOSTED DECISION TREES (GBDT)

GBDT represents a popular model for iid (non-graph) tabular data (Friedman, 2001), whereby remarkably accurate predictions are produced by an ensemble of weak learners. Formally, at each iteration $t$ of boosting, the current ensemble model $f^{(t)}(\boldsymbol{x})$ is updated in an additive manner via

$$f^{(t+1)}(\boldsymbol{x}) = f^{(t)}(\boldsymbol{x}) + \eta^{(t)} h^{(t)}(\boldsymbol{x}),$$

where $h^{(t)}(\boldsymbol{x})$ is a weaker learner selected from some candidate function space $\mathcal{H}$ (typically decision trees), and $\eta^{(t)}$ is the learning rate calculated with the aid of line search. The weak learner $h^{(t)} \in \mathcal{H}$ is chosen to approximate the pseudo-residuals given by the negative gradient of some loss function $\ell$ w.r.t the current model's predictions. This involves solving

$$h^{(t)} = \arg \min_{h \in \mathcal{H}} \sum_{i=1}^{m} \left\| -\frac{\partial \ell(\boldsymbol{y}_i, f^{(t)}(\boldsymbol{x}_i))}{\partial f^{(t)}(\boldsymbol{x}_i)} - h(\boldsymbol{x}_i) \right\|_2^2, \tag{1}$$

where $\{\boldsymbol{x}_i, \boldsymbol{y}_i\}_{i=1}^{m}$ is a set of $m$ training points. Decision trees are constructed by a recursive partition of feature space into $J_t$ disjoint regions $\mathcal{R}_{1t}, \ldots, \mathcal{R}_{J_t t}$ to minimize the loss function (1) and predict a constant value $b_{jt}$ in each region $\mathcal{R}_{jt}$. The output of $h^{(t)}(\boldsymbol{x})$ can be written as the sum: $h^{(t)}(\boldsymbol{x}) = \sum_{j=1}^{J_t} b_{jt} \mathbf{1}_{\mathcal{R}_{jt}}(\boldsymbol{x})$, where $\mathbf{1}$ is the indicator notation. Many efficient GBDT implementations are available today (Ke et al., 2017; Prokhorenkova et al., 2018; Wen et al., 2019),

and these models can easily be trained for nonstandard prediction tasks via custom loss functions (Elsayed et al., 2021; Li et al., 2007; Velthoen et al., 2021).

## 2.2 PRIOR ATTEMPTS TO COMBINE BOOSTING WITH GRAPH NEURAL NETWORKS

The AdaGCN model from Sun et al. (2019) proposes a GNN architecture motivated by the structure of AdaBoost iterations; however, this approach is not actually designed to handle tabular data. More related to our work is the boosted graph neural network (BGNN) approach from Ivanov & Prokhorenkova (2021), which proposes a novel architecture that jointly trains GBDT and GNN models. Within this framework, GBDT extracts predictive information from node features and the GNN accounts for the graph structure, achieving a significant performance increase on various graph datasets with tabular features.

In the first iteration, BGNN builds a GBDT model $f^{(1)}(\boldsymbol{x})$ over the training node features that are treated essentially as iid tabular data. Using all GBDT predictions, BGNN then concatenates these with the original node features and subsequently uses these as inputs to a GNN model. Next, the GNN is trained with $l$ steps of gradient descent to optimize network parameters as well as the input node features themselves by backpropagating gradients entirely through the GNN into its input space. BGNN uses the difference between the optimized node features and input node features as a new target value for the next decision tree in the subsequent boosting round. After GBDT has been updated with the addition of the new tree, it makes predictions that are used to augment the node features for subsequent use in the GNN, and the process repeats until some stopping criteria is met.

While BGNN has thus far demonstrated promising empirical results, there is no guarantee of convergence or even cost function descent. Our EBBS model addresses this issue through a fully integrated alternative described in Section 3, which enjoys convergence guarantees provided in Section 4. Besides its favorable analytical properties, EBBS naturally supports a suite of graph-based regularizers promoting different properties described in Section 3.4, empirically achieves higher accuracy, and is less prone to overfitting without careful hyperparameter-tuning since the graph is accounted for via a simple regularizer instead of an additional parameterized GNN model. Appendix B provides further discussion about the connection between EBBS and BGNN, highlighting key technical differences.

## 3 END-TO-END INTEGRATION OF GRAPH PROPAGATION AND BOOSTING

In this section we describe our method for combining GBDT with graph-aware propagation layers. We first describe a general family of propagation layers that mimic gradient descent steps along a principled graph-regularized loss, followed by the derivation of a bilevel optimization algorithm that exploits these layers. In terms of notation, we will henceforth use $\boldsymbol{m}_i$ to reference the $i$-th row of an arbitrary matrix $\boldsymbol{M}$.

### 3.1 GRAPH-AWARE PROPAGATION LAYERS INSPIRED BY GRADIENT DESCENT

Recently there has been a surge of interest in GNN architectures with layers defined with respect to the minimization of a principled class of graph-regularized energy functions (Klicpera et al., 2018; Ma et al., 2020; Pan et al., 2021; Yang et al., 2021; Zhang et al., 2020; Zhu et al., 2021). In this context, the basic idea is to associate each descent step along an optimization trajectory (e.g., a gradient descent step, power iteration, or related), with a GNN layer, such that in aggregate, the forward GNN pass can be viewed as minimization of the original energy. Hence GNN training can benefit from the inductive bias afforded by energy function minimizers (or close approximations thereof) whose specific form can be controlled by trainable parameters.

The most common instantiation of this framework, with roots in Zhou et al. (2004), begins with the energy

$$\ell_Z(\boldsymbol{Z}) \triangleq \|\boldsymbol{Z} - f(\boldsymbol{X};\boldsymbol{\theta})\|_{\mathcal{F}}^2 + \lambda \operatorname{tr}\left[\boldsymbol{Z}^\top \boldsymbol{L} \boldsymbol{Z}\right], \tag{2}$$

where $\lambda$ is a trade-off parameter, $\boldsymbol{Z} \in \mathbb{R}^{n \times d}$ is a learnable embedding of $d$-dimensional features across $n$ nodes, and $f(\boldsymbol{X};\boldsymbol{\theta})$ denotes a base model (parameterized by $\boldsymbol{\theta}$) that computes an initial target embedding based on the $d_0$-dimensional node features $\boldsymbol{X} \in \mathbb{R}^{n \times d_0}$. We also define the graph Laplacian of $\mathcal{G}$ as $\boldsymbol{L} \in \mathbb{R}^{n \times n}$, meaning $\boldsymbol{L} = \boldsymbol{D} - \boldsymbol{A}$, where $\boldsymbol{D}$ represents the degree matrix.

Intuitively, solutions of (2) reflect a balance between proximity to $f(\boldsymbol{X}; \boldsymbol{\theta})$ and minimal quadratic differences across graph edges as enforced by the trace term $\text{tr}\left[\boldsymbol{Z}^{\top}\boldsymbol{L}\boldsymbol{Z}\right] = \sum_{\{i,j\}\in\mathcal{E}}\|\boldsymbol{z}_i - \boldsymbol{z}_j\|_2^2$; more general regularization factors are considered in Section 3.4. On the positive side, (2) can be solved in closed-form via

$$\widetilde{f}^*(\boldsymbol{X}; \boldsymbol{\theta}) \triangleq \arg\min_{\boldsymbol{Z}} \ell_Z(\boldsymbol{Z}) = \boldsymbol{P}^* f(\boldsymbol{X}; \boldsymbol{\theta}), \quad \text{with } \boldsymbol{P}^* \triangleq (\boldsymbol{I} + \lambda\boldsymbol{L})^{-1}. \tag{3}$$

However, for large graphs the requisite inverse is not practically-feasible to compute, and instead iterative approximations are preferable. To this end, we may initialize as $\boldsymbol{Z}^{(0)} = f(\boldsymbol{X}; \boldsymbol{\theta})$, and then proceed to iteratively descend in the direction of the negative gradient. Given that

$$\frac{\partial \ell_Z(\boldsymbol{Z})}{\partial \boldsymbol{Z}} = 2\lambda\boldsymbol{L}\boldsymbol{Z} + 2\boldsymbol{Z} - 2f(\boldsymbol{X}; \boldsymbol{\theta}), \tag{4}$$

the $k$-th iteration of gradient descent becomes

$$\boldsymbol{Z}^{(k)} = \boldsymbol{Z}^{(k-1)} - \alpha\left[(\lambda\boldsymbol{L} + \boldsymbol{I})\boldsymbol{Z}^{(k-1)} - f(\boldsymbol{X}; \boldsymbol{\theta})\right], \tag{5}$$

where $\frac{\alpha}{2}$ serves as the effective step size. Given that $\boldsymbol{L}$ is generally sparse, computation of (5) can leverage efficient sparse matrix multiplications, and we may also introduce modifications such as Jacobi preconditioning to speed convergence (Axelsson, 1996; Yang et al., 2021). Furthermore, based on well-known properties of gradient descent, if $k$ is sufficiently large and $\alpha$ is small enough, then

$$\widetilde{f}^*(\boldsymbol{X}; \boldsymbol{\theta}) \approx \widetilde{f}^{(k)}(\boldsymbol{X}; \boldsymbol{\theta}) \triangleq \boldsymbol{P}^{(k)}[f(\boldsymbol{X}; \boldsymbol{\theta})], \tag{6}$$

where the operator $\boldsymbol{P}^{(k)}(\cdot)$ computes $k$ gradient steps via (5). The structure of these propagation steps, as well as related variants based on normalized modifications of gradient descent, equate to principled GNN layers, such as those used by GCN (Kipf & Welling, 2016), APPNP (Klicpera et al., 2018), and many others, which can be trained within a broader bilevel optimization framework as described next.

## 3.2 From Graph-Aware Propagation to Bilevel Optimization

Given that the updated embeddings $\widetilde{f}^{(k)}(\boldsymbol{X}; \boldsymbol{\theta})$ represent a graph regularized version of the base estimator $f(\boldsymbol{X}; \boldsymbol{\theta})$, we obtain a natural candidate predictor for application to downstream tasks such as node classification. Of course this presupposes that the parameters $\boldsymbol{\theta}$ can be suitably trained, particularly in an end-to-end manner that accounts for the propagation operator $\boldsymbol{P}^{(k)}$. To this end, we can insert $\widetilde{f}^{(k)}(\boldsymbol{X}; \boldsymbol{\theta})$ within an application-specific meta-loss given by

$$\ell_{\theta}^{(k)}(\boldsymbol{\theta}) \triangleq \sum_{i=1}^{m} \mathcal{D}\bigg(g\left[\widetilde{f}^{(k)}(\boldsymbol{X}; \boldsymbol{\theta})_i\right], \boldsymbol{y}_i\bigg), \tag{7}$$

where $g : \mathbb{R}^d \to \mathbb{R}^c$ is some differentiable node-wise function, possibly an identity mapping, with $c$-dimensional output. Additionally, $\widetilde{f}^{(k)}(\boldsymbol{X}; \boldsymbol{\theta})_i$ is the $i$-th row of $\widetilde{f}^{(k)}(\boldsymbol{X}; \boldsymbol{\theta})$, $m < n$ is the number of labeled nodes and $\mathcal{D}$ is some discriminator function, e.g., cross-entropy for classification, squared error for regression. We may then optimize $\ell_{\theta}^{(k)}(\boldsymbol{\theta})$ over $\boldsymbol{\theta}$ to obtain our final predictive model.

In prior work (Klicpera et al., 2018; Ma et al., 2020; Pan et al., 2021; Yang et al., 2021; Zhang et al., 2020; Zhu et al., 2021), this type of bilevel optimization framework has been adopted to either unify and explain existing GNN models, or motivate alternatives by varying the structure of $\boldsymbol{P}^{(k)}$. However, in all cases to date that we are aware of, it has been assumed that $f(\boldsymbol{X}; \boldsymbol{\theta})$ is differentiable, typically either a linear function or an MLP. Hence because $\boldsymbol{P}^{(k)}$ is also differentiable, $\partial\ell_{\theta}^{(k)}(\boldsymbol{\theta})/\partial\boldsymbol{\theta}$ can be computed to facilitate end-to-end training via stochastic gradient descent (SGD). In contrast, to accommodate tabular data, we instead choose to define $f(\boldsymbol{X}; \boldsymbol{\theta})$ via decision trees such that SGD is no longer viable . Consequently, we turn to gradient boosting as outlined next.

## 3.3 Bilevel Boosting

Let $\boldsymbol{\theta}^{(t)}$ denote the gradient boosting parameters that have accumulated through iteration $t$ such that $f(\boldsymbol{X}; \boldsymbol{\theta}^{(t)})$ represents the weighted summation of weak learners from $1, \ldots, t$. We also assume that

$f(\boldsymbol{X}; \boldsymbol{\theta}^{(t)})$ is separable across nodes, i.e. rows of $\boldsymbol{X}$, such that (with some abuse of notation w.r.t. the domain of $f$) we have $f(\boldsymbol{X}; \boldsymbol{\theta}^{(t)})_i \equiv f(\boldsymbol{x}_i; \boldsymbol{\theta}^{(t)})$. The next gradient boosting iteration proceeds by computing the function-space gradient $\boldsymbol{R}^{(t)} \triangleq$

$$
\frac{\partial \ell_\theta^{(k)}(\boldsymbol{\theta}^{(t)})}{\partial f(\boldsymbol{X}; \boldsymbol{\theta}^{(t)})} = \frac{\partial \ell_\theta^{(k)}(\boldsymbol{\theta}^{(t)})}{\partial \widetilde{f}^{(k)}(\boldsymbol{X}; \boldsymbol{\theta}^{(t)})} \frac{\partial \widetilde{f}^{(k)}(\boldsymbol{X}; \boldsymbol{\theta}^{(t)})}{\partial f(\boldsymbol{X}; \boldsymbol{\theta}^{(t)})} = \frac{\partial \ell_\theta^{(k)}(\boldsymbol{\theta}^{(t)})}{\partial \widetilde{f}^{(k)}(\boldsymbol{X}; \boldsymbol{\theta}^{(t)})} \frac{\partial \boldsymbol{P}^{(k)} \left[ f\left(\boldsymbol{X}; \boldsymbol{\theta}^{(t)}\right) \right]_{1:m}}{\partial f\left(\boldsymbol{X}; \boldsymbol{\theta}^{(t)}\right)},
\tag{8}
$$

where $\boldsymbol{P}^{(k)} \left[ f\left(\boldsymbol{X}; \boldsymbol{\theta}^{(t)}\right) \right]_{1:m}$ refers to the first $m$ rows of $\boldsymbol{P}^{(k)} \left[ f\left(\boldsymbol{X}; \boldsymbol{\theta}^{(t)}\right) \right]$. For illustration purposes, consider the special case where $\mathcal{D}$ is the squared error and $g$ is an identity operator, in which case (8) reduces to

$$
\boldsymbol{R}^{(t)} = 2 \sum_{i=1}^m \left( \widetilde{f}^{(k)}(\boldsymbol{X}; \boldsymbol{\theta}^{(t)})_i - \boldsymbol{y}_i \right) \frac{\partial \boldsymbol{P}^{(k)} \left[ f\left(\boldsymbol{X}; \boldsymbol{\theta}^{(t)}\right) \right]_i}{\partial f\left(\boldsymbol{X}; \boldsymbol{\theta}^{(t)}\right)},
\tag{9}
$$

which as $k \to \infty$, will simplify to

$$
\boldsymbol{R}^{(t)} = 2 \sum_{i=1}^m \left( \widetilde{f}^*(\boldsymbol{X}; \boldsymbol{\theta}^{(t)})_i - \boldsymbol{y}_i \right) (\boldsymbol{p}_i^*)^\top.
\tag{10}
$$

This can be viewed as a graph-aware, smoothed version of the pseudo-residuals obtained from standard gradient boosting applied to the squared-error loss. Next, we fit the best weak learner to approximate these gradients by minimizing the quadratic loss

$$
h^{(t)} = \arg \min_{h \in \mathcal{H}} \sum_{i=1}^n \left\| \boldsymbol{r}_i^{(t)} - h(\boldsymbol{x}_i) \right\|_2^2
\tag{11}
$$

over all nodes $n$. Note that unlike in the canonical gradient boosting setting where function-space gradients are only needed over the training samples, *for EBBS we must fit the weak-learners to gradients from both the training and test nodes.*[1] This is because graph propagation allows the base predictor from a test node to influence the smoothed prediction applied to the labeled training nodes. Finally, we update the base predictor model by adding a new weak learner $h^{(t)}$ via

$$
f\left(\boldsymbol{x}_i; \boldsymbol{\theta}^{(t+1)}\right) = f\left(\boldsymbol{x}_i; \boldsymbol{\theta}^{(t)}\right) + \eta^{(t)} h^{(t)}(\boldsymbol{x}_i) \ \ \forall i = 1, \ldots, n,
\tag{12}
$$

where $\eta^{(t)}$ is the learning rate chosen by line search and $\boldsymbol{\theta}^{(t+1)} = \left\{ \boldsymbol{\theta}^{(t)}, h^{(t)}, \eta^{(t)} \right\}$. The iterative steps of EBBS are summarized in Algorithm 1.

---

**Algorithm 1:** Efficient Bilevel Boosted Smoothing (EBBS)

---

**Input:** Feature matrix $\{\boldsymbol{X}\}$ and target $\{\boldsymbol{y}_i\}_{\{i=1,\ldots,m\}}$;
Initialize $f(\boldsymbol{X}; \boldsymbol{\theta}^{(0)}) = 0$ ;
**for** $t = 0, 1, \ldots, T$ **do**
  Propagate over the graph to compute $\widetilde{f}^{(k)}(\boldsymbol{X}; \boldsymbol{\theta}^{(t)})$ via (6) ;
  Compute the function-space gradient (pseudo-residual) $\boldsymbol{R}^{(t)}$ using (8);
  Execute (11) to find the best weak-learner $h^{(t)}$ ;
  Update the new base model $f(\boldsymbol{X}; \boldsymbol{\theta}^{(t+1)})$ via (12) ;
**end**
**Output:** $\widetilde{f}^{(k)}(\boldsymbol{X}; \boldsymbol{\theta}^{(T)})$;

---

---

[1]This is also one of the key differences between EBBS and BGNN, in that the latter does not fit the gradients that originate from test nodes; see supplementary for details.

### 3.4 BROADER FORMS OF PROPAGATION

**Kernelized EBBS.** Kernel methods have powered traditional machine learning by modeling nonlinear relations among data points by (implicitly) mapping them to a high-dimensional space (Schölkopf & Smola, 2002). Laplacian kernels constitute the graph counterpart of popular translation-invariant kernels in Euclidean space (Smola & Kondor, 2003). The Laplacian kernel matrix is defined as $\boldsymbol{K} := r^\dagger(\boldsymbol{L})$, where $\dagger$ denotes the psedoinverse, and the choice of matrix-valued function $r(\cdot)$ determines which kernel is being used (and which properties of node features are being promoted (Ioannidis et al., 2018)). With a chosen Laplacian kernel, the objective in (2) can be rewritten more generally as

$$\ell_Z(\boldsymbol{Z}) \triangleq \|\boldsymbol{Z} - f(\boldsymbol{X};\boldsymbol{\theta})\|_{\mathcal{F}}^2 + \lambda \mathrm{tr}\left[\boldsymbol{Z}^\top \boldsymbol{K}^\dagger \boldsymbol{Z}\right]. \tag{13}$$

This allows EBBS to utilize more flexible regularizers than (2), such as the diffusion kernel (Smola & Kondor, 2003), where matrix inversion can be avoided by noticing that $\mathrm{tr}[\boldsymbol{Z}^\top \boldsymbol{K}^\dagger \boldsymbol{Z}] = \mathrm{tr}[\boldsymbol{Z}^\top r(\boldsymbol{L})\boldsymbol{Z}]$. EBBS (Algorithm 1) can thus be applied with any positive semi-definite graph kernel and our subsequent convergence results (Theorem 1) will still hold.

**Adaptations for Heterophily Graphs.** For heterophily graphs (Zhu et al., 2020a;b), quadratic regularization may be overly sensitive to spurious edges, so we can adopt more general energy functions of the form

$$\ell_Z(\boldsymbol{Z}) \triangleq \|\boldsymbol{Z} - f(\boldsymbol{X};\boldsymbol{\theta})\|_{\mathcal{F}}^2 + \lambda \sum_{\{i,j\}\in\mathcal{E}} \rho\left(\|\boldsymbol{z}_i - \boldsymbol{z}_j\|_2^2\right). \tag{14}$$

as proposed in Yang et al. (2021). In this expression $\rho : \mathbb{R}^+ \to \mathbb{R}$ is some potentially nonlinear function, typically chosen to be concave and non-decreasing so as to contribute robustness w.r.t edge uncertainty. The resulting operator $\boldsymbol{P}^{(k)}$ can then be derived using iterative reweighted least squares, which results in a principled form of graph attention.

**Label-Aware Error Smoothing.** Huang et al. (2020a) have shown that the prediction errors of a fixed MLP (or related model) can be smoothed via a regularized loss similar to (2), and this produces competitive node classification accuracy. The key difference is to replace the base model $f(\boldsymbol{X};\boldsymbol{\theta})$ in (2) with the error $\boldsymbol{E} \triangleq \boldsymbol{M} \odot (\boldsymbol{Y} - f(\boldsymbol{X};\boldsymbol{\theta}))$, where $\boldsymbol{Y} \in \mathbb{R}^{n \times c}$ is a matrix of labels and $\boldsymbol{M}$ is a mask with $i$-th row equal to ones if $v_i \in \mathcal{L}$ and zero otherwise. The optimal revised solution of (2) then becomes $\boldsymbol{P}^* \boldsymbol{E}$ analogous to (3), with efficient approximation $\boldsymbol{P}^{(k)}[\boldsymbol{E}]$ that spreads the original errors across graph edges. To update the final predictions, this smoothed error is added back to the original base model leading to the final smoothed estimator

$$\widetilde{f}(\boldsymbol{X};\boldsymbol{\theta}) = f(X;\boldsymbol{\theta}) + \boldsymbol{P}^{(k)}\left[\boldsymbol{M} \odot (\boldsymbol{Y} - f(\boldsymbol{X};\boldsymbol{\theta}))\right]. \tag{15}$$

While Huang et al. (2020a) treat this estimator as a fixed postprocessing step, it can actually be plugged into a meta-loss for end-to-end EBBS optimization just as before (per the analysis in Section 3.2 and the differentiability of (15) w.r.t. $f$). Although a potential limitation of this strategy is that the label dependency could increase the risk of overfitting, various counter-measures can be introduced to mitigate this effect (Wang et al., 2021). Moreover, this approach cannot be extended to inductive (or unsupervised) settings where no labels are available at test time.

## 4 CONVERGENCE ANALYSIS OF EBBS

We next consider the convergence of Algorithm 1 given that generally speaking, methods with guaranteed convergence tend to be more reliable, reproducible, and computationally efficient. Let $\ell_\theta^*(\boldsymbol{\theta})$ denote the loss from (7) with $\boldsymbol{P}^{(k)}$ set to $\boldsymbol{P}^*$ such that $\widetilde{f}^{(k)}$ becomes $\widetilde{f}^*$ (i.e., as $k \to \infty$ with $\alpha$ sufficiently small). We further define the optimal boosting parameters applied to this loss as

$$\boldsymbol{\theta}^* \triangleq \arg\min_{\boldsymbol{\theta}\in\boldsymbol{\Theta}} \ell_\theta^*(\boldsymbol{\theta}), \tag{16}$$

where $\boldsymbol{\Theta}$ denotes the space of possible parameterizations as determined by the associated family of weak learners $\mathcal{H}$. In analyzing the convergence of EBBS, we seek to bound the gap between the value of the ideal loss $\ell_\theta^*(\boldsymbol{\theta}^*)$, and the approximation $\ell_\theta^{(k)}(\boldsymbol{\theta}^{(t)})$ that is actually achievable for finite values of $k$ and $t$. In doing so, we must account for the interplay between three factors:

1. The embedding-space gradients from the lower-level loss $\ell_z(\boldsymbol{Z})$ that form $\boldsymbol{P}^{(k)}$.

2. The higher-level function-space gradients $\boldsymbol{R}^{(t)}$ that are endemic to any GBDT-based method, but that in our situation uniquely depend on *both* training and test samples because of the graph propagation operator $\boldsymbol{P}^{(k)}$.

3. The approximation error introduced by the limited capacity of each weak learner that is tasked with fitting the aforementioned function-space gradients.

We explicitly account for these factors via the following result, which establishes convergence guarantees w.r.t. both $k$ and $t$. Note that to enhance the convergence rate w.r.t. $t$, we apply Nesterov's acceleration method (Nesterov, 2003) to each boosting iteration as proposed in Lu et al. (2020).

**Theorem 1.** *Let $\mathcal{D}(\boldsymbol{u}, \boldsymbol{v}) = \|\boldsymbol{u} - \boldsymbol{v}\|_2^2$, $g(\boldsymbol{u}) = \boldsymbol{u}$, and $\alpha = \|\lambda \boldsymbol{L} + \boldsymbol{I}\|_2^{-1}$ in the loss from (7). Then for all $k \geq 0$ and $t \geq 0$, there exists a constant $\Gamma \in (0, 1]$ and $c > 0$ such that the iterations of Algorithm 1, augmented with the momentum factor from Lu et al. (2020) and associated momentum parameter $\phi \leq \frac{\Gamma^4}{4 + \Gamma^2}$, satisfy*

$$\left| \ell_\theta^*(\boldsymbol{\theta}^*) - \ell_\theta^{(k)}(\boldsymbol{\theta}^{(t)}) \right| \leq O\left(\tfrac{1}{t^2} + e^{-ck}\right). \tag{17}$$

The proof, which builds upon ideas from Lu & Mazumder (2020), is deferred to the supplementary.

**Remark 1.** *The quantity $\Gamma$ represents what is referred to as the minimum cosine angle or MCA (as proposed in Lu et al. (2020) and modified in Lu & Mazumder (2020)), which relates to the minimum angle between any candidate residual and the best-fitting weak learner. For example, for strong learners the MCA will be near one, and we can execute near exact gradient descent in functional space. Please see supplementary for further details.*

**Remark 2.** *While Theorem 1 is currently predicted on a quadratic meta-loss, this result can likely be extended to other convex (but not necessarily strongly convex) alternatives, albeit with possibly weaker convergence rates. In contrast, BGNN is predicated on a highly-nonconvex GNN-based loss, and similar convergence guarantees are unlikely to exist. This is especially true given that the function-space gradients used by BGNN are only computed on the training set, such that there is no assurance that each boosting step will not increase the final loss. More specifically, the perturbation of test point embeddings that feed into the subsequent GNN layers (as introduced with each BGNN boosting iteration) is not taken into account. Please see the supplementary for further details regarding the differences between BGNN and EBBS.*

**Remark 3.** *Theorem 1 also holds when the meta-loss from (7) is defined w.r.t. any positive semi-definite kernel from Section 3.4 or the predictor from (15). In contrast, the situation is more nuanced when applying a nonconvex regularizer as in (14) (see supplementary for further details).*

**Corollary 1.** *Let $\mathcal{D}(\boldsymbol{u}, \boldsymbol{v}) = \|\boldsymbol{u} - \boldsymbol{v}\|_2^2$, $g(\boldsymbol{u}) = \boldsymbol{u}$, $\alpha = \|\boldsymbol{L} + \lambda \boldsymbol{I}\|_2^{-1}$ in the loss from (7). Additionally, a penalty factor $\varepsilon \|f(\boldsymbol{X}; \boldsymbol{\theta})\|_{\mathcal{F}}^2$ is added to (7). Then for all $k \geq 0$ and $t \geq 0$, there exists a constant $\beta \in [0, 1)$ and $c > 0$ such that the iterations of Algorithm 1 are such that*

$$\left| \ell_\theta^*(\boldsymbol{\theta}^*) - \ell_\theta^{(k)}(\boldsymbol{\theta}^{(t)}) \right| \leq O\left(\beta^t + e^{-ck}\right). \tag{18}$$

**Remark 4.** *The quantity $\beta$ is related to the density of weak learners in the function space of predictors $f$ and the property of bilevel loss (7). In particular, $\beta = 1 - \frac{\epsilon}{\sigma}\Gamma^2$, where $\Gamma$ is the MCA as defined in Lu & Mazumder (2020), $\sigma$ is a finite value related to the largest singular value of $\boldsymbol{P}$, which can be found in supplementary.*

## 5 EXPERIMENTS

This section describes the datasets, baseline models and experimental details, followed by comparative empirical results.

**Setup.** To study the empirical effectiveness of our methods, we apply them in numerous node regression and classification tasks and compare with various tabular and GNN models. Generally, which class of method fares best will depend on the particular properties of a graph dataset (Faber et al., 2021; Huang et al., 2020a). Our benchmarks specifically focus on graph datasets with rich

| Method | Regression | | | | Classification | | | |
|---|---|---|---|---|---|---|---|---|
| | **House** | **County** | **VK** | **Avazu** | **Slap** | **DBLP** | **CS** | **Phy** |
| **GAT** | 0.54 | 1.45 | 7.22 | 0.1134 | 80.1 | 80.2 | 91.6 | 95.4 |
| **GCN** | 0.63 | 1.48 | 7.25 | 0.1141 | 87.8 | 42.8 | 92.3 | 95.4 |
| **AGNN** | 0.59 | 1.45 | 7.26 | 0.1134 | 89.2 | 79.4 | 92.7 | **96.9** |
| **APPNP** | 0.69 | 1.50 | 13.23 | 0.1127 | 89.5 | 83.0 | 93.2 | 96.6 |
| **CatBoost** | 0.63 | 1.39 | 7.16 | 0.1172 | **96.3** | **91.3** | 91.9 | 94.6 |
| **CatBoost+** | 0.54 | 1.25 | 6.96 | 0.1083 | 96.2 | 90.7 | 94.6 | 96.4 |
| **BGNN** | 0.50 | 1.26 | 6.95 | 0.1090 | 95.0 | 88.9 | 92.5 | 96.4 |
| **EBBS (ours)** | **0.45** | **1.11** | **6.90** | **0.1062** | **96.3** | **91.3** | **94.9** | **96.9** |

Table 1: Root mean squared error (RMSE) of different methods for node regression; accuracy (%) of different methods for node classification. Top results are boldfaced, all of which are statistically significant. Please see supplementary for standard errors and further details, as well as Figure 1 below. Additionally, the BGNN model results are based on conducting a separate hyperparameter sweep for every data set and every random seed. In contrast, EBBS results are based on fixed parameters across random seeds and are mostly shared across datasets.

node features, which entail an important class of applications. For node regression, we consider four real-word graph datasets adopted from Ivanov & Prokhorenkova (2021): House, County, VK, and Avazu, with different types of data for node features. For node classification, we consider two more datasets from Ivanov & Prokhorenkova (2021): SLAP and DBLP, which stem from heterogeneous information networks (HIN). We also consider the Coauthor-CS and Coauthor-Phy datasets (Shchur et al., 2018), which are from the KDD Cup 2016 challenge and based on the Microsoft Academic Graph. We chose these because they involve relatively large graphs with original sparse features, as opposed to low-dimensional projections (e.g. of originally text features) that are often better handled with just regular MLP-based GNNs or related.[2] For contrast though, we also include OGB-ArXiv (Hu et al., 2020) with low-dimensional homogeneous node features favorable to GNNs.

We compare our proposed methods against various alternatives, starting with purely tabular baselines in which the graph structure is ignored and models are fit to node features as if they were iid. Here we consider **CatBoost**, a high-performance GBDT implementation with sophisticated strategies for handling categorical features and avoiding overfitting (Prokhorenkova et al., 2018). We also evaluate popular GNN models: **GCN** (Kipf & Welling, 2016), **GAT** (Veličković et al., 2017), **AGNN** (Thekumparampil et al., 2018) and **APPNP** (Klicpera et al., 2018). Finally, we compare against the hybrid approach, **BGNN** (Ivanov & Prokhorenkova, 2021), which combines GBDT and GNN via end-to-end training and represents the current SOTA for boosting-plus-tabular architectures for graph data. Our specific GBDT implementation of EBBS is based on top of CatBoost, where we interleave the boosting rounds of CatBoost with our proposed propagation steps that define the effective loss used for training. The supplementary contains comprehensive details about our datasets, models/hyperparameters, and overall experimental setup.

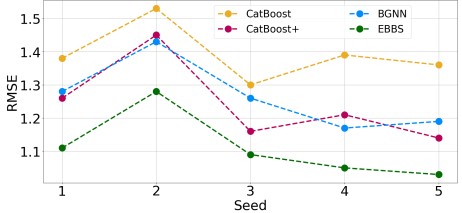

Figure 1: RMSE v.s. random seed for County dataset.

| OGB-ArXiv | | | | | |
|---|---|---|---|---|---|
| Method | **MLP** | **CatBoost** | **BGNN** | **EBBS** | **SOTA** |
| Accuracy | 55.50 | 51.0 | 67.0 | 70.10 | 74.31 |

Table 2: Accuracy results on the OGB-ArXiv dataset. As mentioned in the text, the OGB-ArXiv node features are *not* generally favorable to boosted tree models. SOTA is taken from the OGB leaderboard.

**Results.** In Table 1 we present the results for the various node regression and node classification benchmarks. Here EBBS outperforms various baselines across datasets. The baseline GNN models are all clearly challenged by the tabular features in these datasets, with the exception of Coauthor-CS and Coauthor-Phy where GNNs perform reasonably well as expected.

---

[2]Although tabular graph data for node classification is widely-available in industry, unfortunately there is currently little publicly-available, real-world data that can be used for benchmarking.

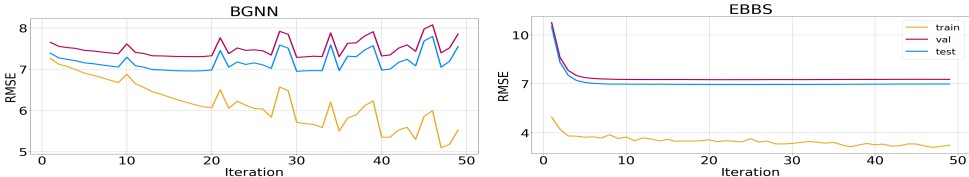

Figure 2: Performance of BGNN vs. EBBS after each boosting iteration on the VK dataset. Each model uses the hyperparameter values that performed best on the County dataset.

Importantly, all bold-faced results reported in Table 1 are significant once we properly account for the trial-to-trial variability induced by the random training splits shared across all methods; see supplementary for full presentation of standard errors and related analysis. Briefly here, for classification results, bold-faced results are statistically significant with respect to standard errors. In contrast, for node regression the standard errors for some datasets are a bit larger relative to the EBBS improvement gap, but this turns out to be a spurious artifact of the shared trial-to-trial variance. More specifically, EBBS actually outperforms all other methods across all trials on all regression benchmarks, such that a stable performance gap is maintained even while the absolute RMSE of methods may vary for different training splits. While full details are deferred to the supplementary, in Figure 1 we plot the RMSE for five random splits of the County dataset, observing that EBBS outperforms the other baselines across every instance.

Additionally, as a natural ablation of our end-to-end bilevel boosting model, we consider fitting GBDT in the usual fashion (to just the node features without edge information) and subsequently applying EBBS-like graph propagation post hoc, only after all GBDT boosting rounds have already finished. We refer to this approach as **CatBoost+**, which can be viewed as EBBS without end-to-end training. Table 1 and Figure 1 show that our EBBS bilevel boosting, with propagation interleaved inside each boosting round, outperforms post-hoc application of the same propagation at the end of boosting. This is not true however for BGNN, which in aggregate performs similarly to CatBoost+.

Note also that for SLAP and DBLP, BGNN is worse than CatBoost, likely because the graph structure is not sufficiently helpful for these data, which also explains why most GNN baselines perform relatively poorly. Nonetheless, EBBS still achieves comparable results with CatBoost, revealing that it may be more robust to non-ideal use cases. Additionally, for Coauthor-CS and Coauthor-Phy, we observe that EBBS produces competitive results supporting the potential favorability of EBBS relative to GNN models when applied to certain graphs with high dimensional and sparse node features.

Proceeding further, given that the OGB-ArXiv node features are homogeneous, low-dimensional embeddings of an original text-based source, which are *not* generally well-suited for GBDT models, we would not expect BGNN or EBBS to match the SOTA achievable by GNNs. This conjecture is supported by the superiority of the MLP baseline over CatBoost in Table 2. Even so, we observe that EBBS is still more robust relative to BGNN in this setting. Finally, to showcase the relative stability of EBBS relative to BGNN, we store the model hyperparameters that performed best on the County dataset and then trained the respective models on VK data. The resulting training curves are shown in Figure 2, where BGNN exhibits some degree of relative instability. This suggests that in new application domains it may conceivably be easier to adapt EBBS models.

## 6 DISCUSSION

This paper considered various forms of graph propagation to facilitate the application of tabular modeling to node prediction tasks in graph-structured data. We developed a simple yet highly accurate model by interleaving propagation steps with boosting rounds in GBDT. And unlike BGNN, which relies on separate boosting and GNN training steps with no unified loss, our method fully integrates graph propagation within the actual definition of a single bi-level boosting objective. This allows us to establish converge guarantees and avoids the complexity of separate trainable GBDT and GNN modules. Note also that our EBBS algorithm is not specific to GBDT, but can also be used to boost arbitrary weak learners including neural networks (Cortes et al., 2017) or heterogeneous collections (Parnell et al., 2020).

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

# Supplementary Materials

## A ASSESSING STATISTICAL SIGNIFICANCE OF RESULTS

Table S1 shows the mean squared error of different methods for node regression with error bars. While the gains of EBBS may appears insignificant relative to the error bars, this is not actually the case upon closer inspection. Because we used shared training and testing splits for different methods, we can compare how much of this variance is merely caused by different splits while the relative performance is preserved. We would like to point out that the error bars shown in Table S1 merely reflect shared trial-to-trial variability that does not significantly influence relative performance. To this end, below we present the results of each trial for all datasets. For each random seed, we use the fixed training and testing split and show the performance for different methods. From Table S2, we observe that EBBS outperforms BGNN across every instance. Regarding the classification task shown in Table S3, EBBS outperforms BGNN by a significant amount even if we take into account the reported error bars.

| | Regression | | | |
|---|---|---|---|---|
| Method | House | County | VK | Avazu |
| **GAT** | $0.54 \pm 0.01$ | $1.45 \pm 0.06$ | $7.22 \pm 0.29$ | $0.1134 \pm 0.01$ |
| **GCN** | $0.63 \pm 0.01$ | $1.48 \pm 0.08$ | $7.25 \pm 0.19$ | $0.1141 \pm 0.02$ |
| **AGNN** | $0.59 \pm 0.01$ | $1.45 \pm 0.08$ | $7.26 \pm 0.20$ | $0.1134 \pm 0.02$ |
| **APPNP** | $0.69 \pm 0.01$ | $1.50 \pm 0.11$ | $13.23 \pm 0.12$ | $0.1127 \pm 0.01$ |
| **CatBoost** | $0.63 \pm 0.01$ | $1.39 \pm 0.07$ | $7.16 \pm 0.20$ | $0.1172 \pm 0.02$ |
| **CatBoost+** | $0.54 \pm 0.01$ | $1.25 \pm 0.11$ | $6.96 \pm 0.21$ | $0.1083 \pm 0.02$ |
| **BGNN** | $0.50 \pm 0.01$ | $1.26 \pm 0.08$ | $6.95 \pm 0.21$ | $0.1090 \pm 0.01$ |
| **EBBS (ours)** | $\mathbf{0.45 \pm 0.01}$ | $\mathbf{1.11 \pm 0.09}$ | $\mathbf{6.90 \pm 0.23}$ | $\mathbf{0.1062 \pm 0.01}$ |

Table S1: Mean squared error of different methods for node regression. Top results are boldfaced.

## B ANALYSIS OF THE DIFFERENCES BETWEEN EBBS AND BGNN

Although EBBS is clearly quite related to BGNN as discussed in the main text, there remain significant differences as follows:

**Integrated Loss:** EBBS integrates both the boosting module and graph propagation within a single unified bi-level objective function, which allows for meaningful convergence guarantees. In contrast, the BGNN boosting rounds are not actually minimizing a unified objective function (due to changing GNN parameters each round), which restricts the possibility of convergence guarantees. In BGNN, the GNN is treated as a trainable module stacked on top of GBDT, and both the GNN and GBDT are updated with their respective losses by iterating back-and-forth during training. In contrast, the EBBS setup is different, with only a single unchanging GBDT-based objective involved at a high level. In fact, we can view even the EBBS graph propagation step as being fully integrated within the actual definition of a single static GBDT training objective. To illustrate this more explicitly, the EBBS objective from

$$\ell_\theta^{(k)}(\boldsymbol{\theta}) \triangleq \sum_{i=1}^{m} \mathcal{D}\bigg(g\left[\widetilde{f}^{(k)}(\boldsymbol{X};\boldsymbol{\theta})_i\right], \boldsymbol{y}_i\bigg), \tag{19}$$

is equivalent to plugging a parameterized GBDT predictive model into the fixed training loss. There is no need to iterate back-and-forth between adding weak learners to $f(\boldsymbol{X};\boldsymbol{\theta})$ and separate GNN parameter updates because all trainable parameters are inside of the function $f(\boldsymbol{X};\boldsymbol{\theta})$, and standard gradient boosting can be used with the above loss. Of course as we know, standard gradient boosting iterations involve fitting function space gradients using a quadratic loss, which is exactly (11) in our paper. And while of course (11) keeps changing as the gradients change, the objective (19) upon which these gradients are based does not. If we reinterpret BGNN from a similar perspective, then

| Regression | | | | | | |
|---|---|---|---|---|---|---|
| Dataset | Method | Seed 1 | Seed 2 | Seed 3 | Seed 4 | Seed 5 |
| **House** | **CatBoost** | 0.62 | 0.63 | 0.62 | 0.63 | 0.64 |
| | **CatBoost+** | 0.53 | 0.55 | 0.53 | 0.54 | 0.56 |
| | **BGNN** | 0.49 | 0.51 | 0.49 | 0.52 | 0.50 |
| | **EBBS** | **0.44** | **0.46** | **0.44** | **0.46** | **0.46** |
| **County** | **CatBoost** | 1.38 | 1.53 | 1.30 | 1.39 | 1.36 |
| | **CatBoost+** | 1.26 | 1.45 | 1.16 | 1.21 | 1.14 |
| | **BGNN** | 1.28 | 1.43 | 1.26 | 1.17 | 1.19 |
| | **EBBS** | **1.11** | **1.28** | **1.09** | **1.05** | **1.03** |
| **VK** | **CatBoost** | 7.13 | 7.29 | 7.48 | 7.20 | 6.84 |
| | **CatBoost+** | 6.88 | 7.07 | 7.24 | 7.01 | 6.60 |
| | **BGNN** | 6.90 | 7.05 | 7.26 | 6.98 | 6.60 |
| | **EBBS** | **6.82** | **7.01** | **7.21** | **6.92** | **6.54** |
| **Avazu** | **CatBoost** | 0.1046 | 0.1057 | 0.1133 | 0.1515 | 0.1111 |
| | **CatBoost+** | 0.0970 | 0.0992 | 0.1057 | 0.1400 | 0.0998 |
| | **BGNN** | 0.0979 | 0.0992 | 0.1082 | 0.1332 | 0.0999 |
| | **EBBS** | **0.0967** | **0.0990** | **0.1045** | **0.1331** | **0.0979** |

Table S2: Mean squared error of different methods for different random seeds, which presents different training and testing splits. Top results are boldfaced.

| Classification | | | | |
|---|---|---|---|---|
| Method | Slap | DBLP | Coauthor-CS | Coauthor-Phy |
| **GAT** | $80.1 \pm 1.0$ | $80.2 \pm 1.0$ | $91.55 \pm 0.5$ | $95.4 \pm 0.1$ |
| **GCN** | $87.8 \pm 1.0$ | $42.8 \pm 4.0$ | $92.32 \pm 0.4$ | $95.4 \pm 0.1$ |
| **AGNN** | $89.2 \pm 1.0$ | $79.4 \pm 1.0$ | $92.65 \pm 0.4$ | $\mathbf{96.9 \pm 0.1}$ |
| **APPNP** | $89.5 \pm 1.0$ | $83.0 \pm 2.0$ | $93.2 \pm 0.1$ | $96.6 \pm 0.2$ |
| **CatBoost** | $\mathbf{96.3 \pm 0.0}$ | $\mathbf{91.3 \pm 1.0}$ | $91.9 \pm 0.2$ | $94.6 \pm 0.2$ |
| **CatBoost+** | $96.2 \pm 0.3$ | $90.7 \pm 1.1$ | $94.6 \pm 0.1$ | $96.4 \pm 0.1$ |
| **BGNN** | $95.0 \pm 0.0$ | $88.9 \pm 1.0$ | $92.5 \pm 0.2$ | $96.4 \pm 0.0$ |
| **EBBS (ours)** | $\mathbf{96.3 \pm 0.4}$ | $\mathbf{91.3 \pm 0.6}$ | $\mathbf{94.9 \pm 0.2}$ | $\mathbf{96.9 \pm 0.2}$ |

Table S3: Accuracy (%) of different methods for node classification. Top results are boldfaced.

each time the GNN parameters are updated, the implicit loss as seen by the BGNN boosting step, analogous to (19) above, would change.

**Incorporating Test Nodes During Training:** a second key difference is the fact that in BGNN, input features of nodes in the test set are only used to predict test node labels for passage to the GNN; they are not used to approximate gradients/residuals evaluated at testing nodes during the selection of GBDT weak learners. In contrast, the GBDT objective function in EBBS itself explicitly depends on test node input features because these features propagate over the graph to influence training node predictions as defined via (19). When training our EBBS model, gradient information from test nodes (or at least those receiving significant signal) must be passed back to the boosting module. Or stated differently, unlike standard boosting as applied to iid training samples, we must predict function-space gradients/residuals across all nodes when adding each new weak learner. This is because the input to the propagation operator $\boldsymbol{P}^{(k)}(\cdot)$ includes features from all nodes (both training and testing), and why (11) involves a summation over all $n$ testing nodes, not just the $m$ training nodes.

**Impact of Multiple SGD Steps:** During each round of BGNN training, multiple SGD steps are applied to both update GNN parameters and GNN inputs (whereas there is no similar operation in EBBS). These multiple inner iterations, while needed for GNN training and the presented empirical

| Regression | | | | |
|---|---|---|---|---|
| Method | House | County | VK | Avazu |
| **BGNN** | $0.50 \pm 0.01$ | $1.26 \pm 0.08$ | $6.95 \pm 0.21$ | $0.1090 \pm 0.01$ |
| **BGNN**(1 SGD step per iter.) | $0.56 \pm 0.01$ | $1.34 \pm 0.09$ | $7.28 \pm 0.23$ | $0.110 \pm 0.02$ |
| **BGNN**(no end-to-end train) | $0.51 \pm 0.01$ | $1.33 \pm 0.08$ | $7.07 \pm 0.20$ | $0.1095 \pm 0.01$ |

Table S4: Ablation study for BGNN with one step of gradient step per iteration.

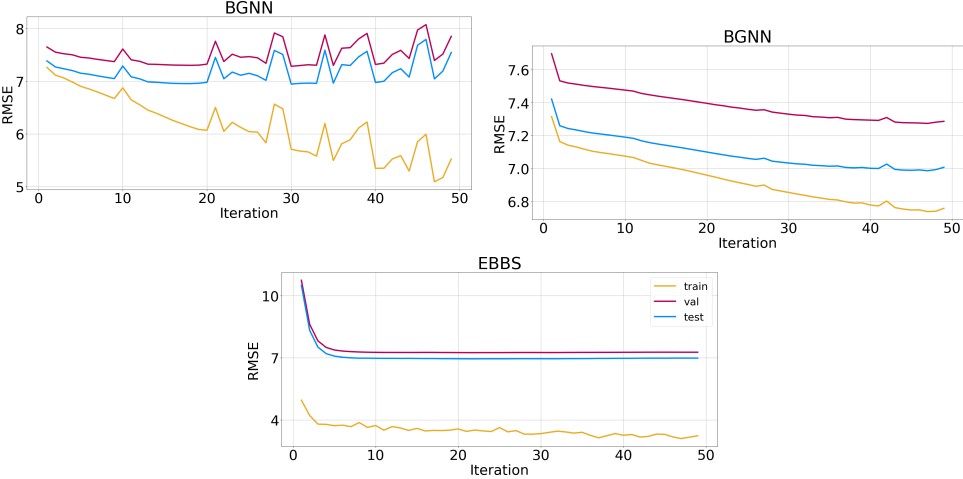

Figure S1: Performance of BGNN vs. EBBS after each boosting iteration on the VK dataset. The left BGNN uses the hyperparameter values that performed best on the County dataset. The right BGNN uses the hyperparameter values that performed best on the VK dataset. EBBS uses the hyperparameter values that performed best on the County dataset. These results suggest that EBBS can be run with mostly shared hyperparameters across all datasets.

results, do not explicitly account for the outer-loop boosting in a fully integrated fashion. While BGNN can be executed with a single SGD step per iteration, this alone is not adequate for establishing convergence and it comes at the cost of decreased empirical performance. Table S4 compares BGNN performance using only a single gradient step per iteration, where results are obtained using a full sweep of all BGNN hyperparameters (as per the authors' published code). These results show that BGNN performance can degrade with only a single SGD step, to the extent that simply training CatBoost plus a separate GNN model on top (i.e., BGNN without end-to-end training, called Res-GNN by Ivanov & Prokhorenkova (2021)) performs just as well, and possibly better on VK.

In part due to the above issues, the BGNN model relies on a grid search over many hyperparameter combinations using each separate validation data set to achieve the results reported by Ivanov & Prokhorenkova (2021). Table S5 and Figure S1 demonstrates that EBBS can be run with mostly shared hyperparameters across all datasets while still performing well.

## C EMPIRICAL CONVERGENCE OF EBBS

In this section, we visualize the losses on the training and validation data over the course of training iterations. Here we consider the County dataset as well as three methods: EBBS, EBBS with momentum and BGNN. For BGNN, we first evaluated different hyperparameter settings on the validation data, and then show the loss curves of the BGNN model with the best hyperparameters. Over each epoch, BGNN performs 10 or 20 steps of gradient descent to train its GNN model. In order to compare our method fairly, we unfold BGNN's inner loop of GNN training, and plot the loss after BGNN has performed one step of gradient descent. Figure S2 shows the loss of BGNN (blue line) oscillates heavily over the course of its iterations. In contrast, the loss of EBBS (pink line and orange lines) decreases monotonically, revealing empirical convergence of EBBS as guaranteed by Theorem

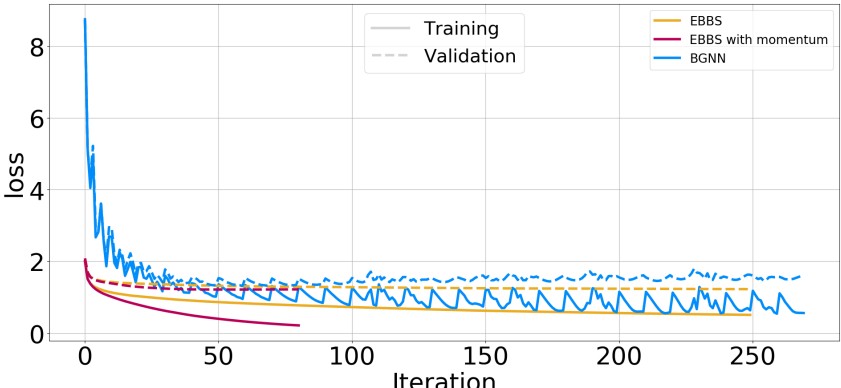

Figure S2: Training and validation losses vs. number of iterations for the County dataset (regression).

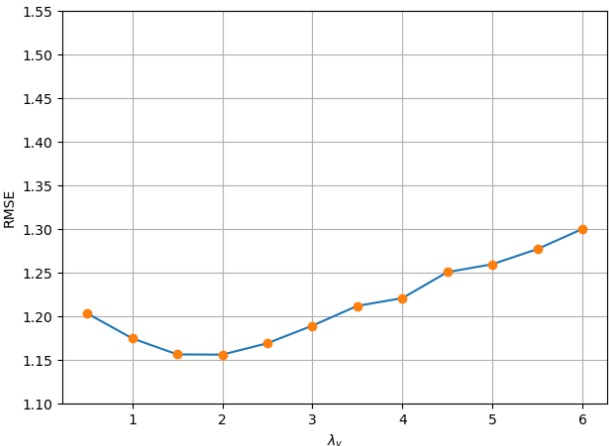

Figure S3: RMSE vs. different value of $\lambda_y$ for County dataset.

1. Comparing EBBS with momentum (pink line) and EBBS without momentum (orange line) reveals that Nesterov's acceleration method can speed up the convergence of our first-order boosting method.

## D  EXPERIMENT DETAILS

### D.1  DATASETS

**House**: node features are the property, edges connect the neighbors and target is the price of the house. **County**: each node is a county and edges connect two counties sharing a border, the target is the unemployment rate for a county. **VK**: each node is a person and edges connect two people based on the friendships, the target is the age of a node. **Avazu**: each node is a device and edges connect two devices if they appear on the same site with the same application, the target is the click-through-rate of a node. **DBLP**: node feature represent authors, paper and conferences and target is to predict one of the classes (database, data mining, information retrieval and machine learning). **SLAP**: node features represent network bioinformatics such as chemical compound, gene and pathway, the goal is to predict one of 15 gene type. **Coauthor**: nodes are authors and an edge connects two authors co-authored a paper. Node features represent paper keywords for each author's papers, and the goal is to predict which fields of study for each author.

For **House**, **County**, **Vk**, **Avazu**, **DBLP** and **SLAP**, Ivanov & Prokhorenkova (2021) introduce how to construct the graph model based on tabular data. Furthermore, we randomly split each dataset (train : validation : test = 6 : 2 : 2), and repeat all experiments with five different splits reporting the

average performance (and its standard deviation) over the five replicates. For fair comparison, we use the same splits and random seed as used for BGNN.

## D.2 Baseline Details

For regression tasks: GNN baselines are from Ivanov & Prokhorenkova (2021).

For classification tasks: GNN baselines for Slap and DBLP are taken from Ivanov & Prokhorenkova (2021). For Coauthor-CS and Coauthor-Phy, we implement the GNN models via the DGL framework (Wang et al., 2019). For GCN, GAT, and AGNN we train 3 layer models with a hidden node representation of 256. APPNP is implemented with 3 layer MLP before the final message passing layer. We use mini-batch training with neighbor sampling, with a fanout of 15 at each layer, to construct the training subgraph for each mini-batch. We do not tune the hyper-parameters for these models. All models are trained with the Adam optimizer with a learning rate of $5e^{-3}$. The results presented for each are the average of 5 runs on the dataset.

## D.3 Model Hyperparameters

We smooth the node features and concatenate the smoothed values with the original feature values for regression task. Here $\lambda_x$ is the weight from (13) and $k_x$ is the number of propagation steps. For all regression models, we choose $\lambda_x = 20$ and $k_x = 5$; for classification we do not use smoothed node features.

**EBBS Model.** We consider the following hyperparamters:

- $\lambda_y, k_y$: We propagate over the graph to compute $\widetilde{f}^{(k)}(\boldsymbol{X}; \boldsymbol{\theta}^{(t)})$ via (6), where $\lambda_y$ is the weight in (2) (or analogously the more general (13)) and $k_y$ is the number of propagation steps.

- *Smoothing Type*: For the graph propagation required to computed $\widetilde{f}^{(k)}(\boldsymbol{X}; \boldsymbol{\theta}^{(t)})$, we choose label smoothing via (6), error smoothing using (15) or both of them.

- *Boosting Tree LR*: The learning rate of our GBDT model.

Additionally, we encode categorical features as numerical variables via the CatBoost Encoder[3] Ivanov & Prokhorenkova (2021). For regression tasks, the number of training epochs is 1000, early stopping rounds is 100. For classification tasks, the number of training epochs is 200, early stopping rounds is 20. Throughout all experiments, we just used the simple Laplacian kernel $\boldsymbol{L}$ for simplicity, and reserve exploration of alternative kernels in diverse settings for future work. For each iteration, we use CatBoost to find the best weak-learner. All hyperparameter settings are listed in Table S5, and were chosen manually based on validation scores in preliminary experiments. We also use an approach from (Wang et al., 2021) to mitigate the data leakage problem.

| Dataset | $\lambda_y$ | $k_y$ | Smoothing Type | Boosting Tree LR |
|---|---|---|---|---|
| House | 2.0 | 5 | label and error | 0.2 |
| County | 2.0 | 5 | label and error | 0.2 |
| VK | 50.0 | 2 | error | 0.05 |
| Avazu | 2.0 | 5 | label and error | 0.1 |
| Classification (all) | 2.0 | 5 | label | 1.0 |

Table S5: Hyperparameter settings used for EBBS, which are shared across all random seeds/dataset-splits. We emphasize here that all BGNN results (the SOTA model for graph data with tabular node features) follow the code from `https://github.com/nd7141/bgnn`, which executes a costly independent sweep over 6 hyperparameters for each dataset and for each random seed/dataset-split to achieve the reported performance.

---

[3]http://contrib.scikit-learn.org/category_encoders/catboost.html

# E  TECHNICAL DETAILS REGARDING CONVERGENCE

To proceed in a quantifiable way, we rely the concept of *minimal cosine angle* (MCA) defined in Lu & Mazumder (2020); Lu et al. (2020).

**Definition 1.** *Given a set of $K$ weak learners, let $\boldsymbol{B} \in \mathrm{R}^{n \times K}$ denote the matrix of predictions of every weak learner over each sample, both training and testing.[4] Then the MCA of this set of weak-learners is defined as*

$$\Gamma \triangleq \min_{\boldsymbol{r} \in \mathbb{R}^n} \max_{j \in [1, \ldots, K]} \cos(\boldsymbol{b}_{:j}, \boldsymbol{r}), \tag{20}$$

*where $\boldsymbol{b}_{:j}$ denotes the $j$-th column of $\boldsymbol{B}$.*

Per this definition, $\Gamma \in (0, 1]$ can be viewed as an estimate of the "density" of the weak learners in the prediction space, with larger values associated with higher density. We also require the definition of $\sigma$-smooth and $\mu$-strongly convex functions:

**Definition 2.** *A loss function $\mathcal{L}(y, f)$ is $\sigma$-smooth if for any $y$ and any $f_1$ and $f_2$, we have that*

$$\mathcal{L}(y, f_1) \leq \mathcal{L}(y, f_2) + \frac{\partial \mathcal{L}(y, f_2)}{\partial f}(f_1 - f_2) + \frac{\sigma}{2}(f_1 - f_2)^2.$$

*Furthermore, $\mathcal{L}(y, f)$ is $\mu$-strongly convex if for any $y$ and any $f_1$ and $f_2$, we have that*

$$\mathcal{L}(y, f_1) \geq \mathcal{L}(y, f_2) + \frac{\partial \mathcal{L}(y, f_2)}{\partial f}(f_1 - f_2) + \frac{\mu}{2}(f_1 - f_2)^2.$$

We then apply these definitions to obtain two intermediate results that will be leveraged for the main proof:

**Lemma 1.** *The energy function (2) is $\sigma$-smooth and $\mu$-strongly convex, with $\sigma = \sigma_{\max}(\lambda \boldsymbol{L} + \boldsymbol{I})$ and $\mu = \sigma_{\min}(\lambda \boldsymbol{L} + \boldsymbol{I})$, where $\sigma_{\max}(\lambda \boldsymbol{L} + \boldsymbol{I})$ and $\sigma_{\min}(\lambda \boldsymbol{L} + \boldsymbol{I})$ represent the largest and smallest singular values of $\lambda \boldsymbol{L} + \boldsymbol{I}$, respectively.*

The proof is straightforward and follows from computing the Hessian of $\ell_Z(\boldsymbol{Z})$. Additionally, since $\boldsymbol{L}$ is the Laplacian of $\mathcal{G}$, it is positive-semidefinite such that we have $0 < \mu \leq \sigma < +\infty$.

We next turn to the meta-loss (7). Note that if $\boldsymbol{Z}^{(0)} = f(\boldsymbol{X}; \boldsymbol{\theta})$, then the operator $\boldsymbol{P}^{(k)}$ collapses to a matrix, meaning $\boldsymbol{P}^{(k)}[f(\boldsymbol{X}; \boldsymbol{\theta})] = \boldsymbol{P}^{(k)} f(\boldsymbol{X}; \boldsymbol{\theta})$, where $\boldsymbol{P}^{(k)} \in \mathbb{R}^{n \times n}$. This observation then leads to the following:

**Lemma 2.** *Let $\mathcal{D}(\boldsymbol{u}) = \|\boldsymbol{u}\|_2^2$, $g(\boldsymbol{u}) = \boldsymbol{u}$, $\alpha = \|\lambda \boldsymbol{L} + \boldsymbol{I}\|_2^{-1}$ in the loss from (7), then we have $\ell_\theta^{(k)}$ is $\sigma^{(k)}$-smooth, with $\sigma^{(k)} = \sigma_{\max}(\boldsymbol{P}_{1:m}^{(k)})$, where $\sigma_{\max}(\boldsymbol{P}_{1:m}^{(k)})$ represents the largest singular values of operator $\boldsymbol{P}_{1:m}^{(k)}$, the subscript $1 : m$ indicates the first $m$ rows or elements of a matrix or vector respectively. Moreover, we have $0 < \sigma^{(k)} < +\infty$ for any $k \geq 0$.*

The proof is also straightforward given that $\boldsymbol{P}^{(k)}$ converges to $(\lambda \boldsymbol{L} + \boldsymbol{I})^{-1}$ for any $\alpha \leq \|\lambda \boldsymbol{L} + \boldsymbol{I}\|_2^{-1}$, and the largest singular value of $\boldsymbol{P}_{1:m}^{(k)}$ is finite.

We now proceed to our specific results related to convergence in the main paper.

**Theorem 1.** *Let $\mathcal{D}(\boldsymbol{u}, \boldsymbol{v}) = \|\boldsymbol{u} - \boldsymbol{v}\|_2^2$, $g(\boldsymbol{u}) = \boldsymbol{u}$, and $\alpha = \|\lambda \boldsymbol{L} + \boldsymbol{I}\|_2^{-1}$ in the loss from (7). Then for all $k \geq 0$ and $t \geq 0$, there exists a constant $\Gamma \in (0, 1]$ and $c > 0$ such that the iterations of Algorithm 1, augmented with the momentum factor from Lu et al. (2020) and associated momentum parameter $\phi \leq \frac{\Gamma^4}{4 + \Gamma^2}$, satisfy*

$$\left| \ell_\theta^*(\boldsymbol{\theta}^*) - \ell_\theta^{(k)}(\boldsymbol{\theta}^{(t)}) \right| \leq O\left(\frac{1}{t^2} + e^{-ck}\right). \tag{21}$$

---

[4]In the original definition, the MCA only concerns training samples; however, for our situation we must consider all samples due to gradient propagation over the whole graph from the perspective of the weak learners in function space.

*Proof.* Given the stated assumptions, and the initialization $\boldsymbol{Z}^{(0)}$ discussed above and in the main text, the loss from (7) reduces to $\ell_\theta^{(k)}(\boldsymbol{\theta}) =$

$$\sum_{i=1}^{m}\left(\widetilde{f}^{(k)}(\boldsymbol{X};\boldsymbol{\theta})_i - \boldsymbol{y}_i\right)^2 = \sum_{i=1}^{m}\left(\boldsymbol{P}^{(k)}[f(\boldsymbol{X};\boldsymbol{\theta})]_i - \boldsymbol{y}_i\right)^2 = \left\|\boldsymbol{P}_{1:m}^{(k)}f(\boldsymbol{X};\boldsymbol{\theta}) - \boldsymbol{y}_{1:m}\right\|_2^2, \quad (22)$$

where the subscript $1:m$ indicates the first $m$ rows or elements of a matrix or vector respectively.

Lemma 2 implies that $\ell_\theta^{(k)}$ is $\sigma^{(k)}$-smooth. We may then apply gradient boosting machine convergence results from Lu et al. (2020)[Theorem 4.1] that apply to this setting. Specifically, assuming a constant step-size rule with $\eta = 1/\sigma^{(k)}$ and momentum parameter $\phi \leq \Gamma^4/(4+\Gamma^2)$, where $\Gamma$ denotes the value of the corresponding MCA defined above, it follows that

$$\ell_\theta^{(k)}(\boldsymbol{\theta}^{(t)}) - \ell_\theta^{(k)}(\boldsymbol{\theta}^*) \leq \frac{\sigma^{(k)}}{2\phi(t+1)^2}\|f(\boldsymbol{X};\boldsymbol{\theta}^*)\|_2^2 \quad (23)$$

for all $t \geq 0$.

Furthermore, based on Bubeck (2014)[Theorem 3.10], Lemma 1, and the step-size $\alpha = 1/\sigma$, we can establish that the descent iterations from (6), which reduce (2), will satisfy

$$\left\|\bar{\boldsymbol{P}}^{(k)}f(\boldsymbol{X};\boldsymbol{\theta}) - \bar{\boldsymbol{P}}^*f(\boldsymbol{X};\boldsymbol{\theta})\right\|_2^2 \leq e^{-\frac{\mu}{\sigma}k}\left\|\bar{\boldsymbol{P}}^{(0)}f(\boldsymbol{X};\boldsymbol{\theta}) - \bar{\boldsymbol{P}}^*f(\boldsymbol{X};\boldsymbol{\theta})\right\|_2^2. \quad (24)$$

where $\bar{\boldsymbol{P}} \triangleq \boldsymbol{P}_{1:m}$ for all variants. Since this bound must hold for any value of $f(\boldsymbol{X};\boldsymbol{\theta})$, by choosing $f(\boldsymbol{X};\boldsymbol{\theta}) = \boldsymbol{I}$, we can infer that

$$\left\|\bar{\boldsymbol{P}}^{(k)} - \bar{\boldsymbol{P}}^*\right\|_2^2 \leq e^{-\frac{\mu}{\sigma}k}\left\|\bar{\boldsymbol{P}}^{(0)} - \bar{\boldsymbol{P}}^*\right\|_2^2, \quad (25)$$

or equivalently

$$\left\|\bar{\boldsymbol{P}}^{(k)} - \bar{\boldsymbol{P}}^*\right\|_2 \leq e^{-\frac{\mu}{2\sigma}k}\left\|\bar{\boldsymbol{P}}^{(0)} - \bar{\boldsymbol{P}}^*\right\|_2. \quad (26)$$

Denote $f^{(k)}(\boldsymbol{X};\boldsymbol{\theta}^*)$ as the optimal solution of $\ell_\theta^{(k)}$. From (23), it follows that

$$\left\|\bar{\boldsymbol{P}}^{(k)}f\left(\boldsymbol{X};\boldsymbol{\theta}^{(t)}\right) - \bar{\boldsymbol{y}}\right\|_2^2 - \left\|\bar{\boldsymbol{P}}^{(k)}f^{(k)}(\boldsymbol{X};\boldsymbol{\theta}^*) - \bar{\boldsymbol{y}}\right\|_2^2 \leq \frac{\sigma^{(k)}}{2\phi(t+1)^2}\|f(\boldsymbol{X};\boldsymbol{\theta}^*)\|_2^2, \quad (27)$$

where $\bar{\boldsymbol{y}} \triangleq \boldsymbol{y}_{1:m}$ analogous to $\bar{\boldsymbol{P}}$. Denote $f(\boldsymbol{X};\boldsymbol{\theta}^*)$ as the optimal solution of $\ell_\theta^*$. Since $f^{(k)}(\boldsymbol{X};\boldsymbol{\theta}^*)$ is the optimal solution of $\ell_\theta^{(k)}$, we also have that

$$\begin{aligned}
\left\|\bar{\boldsymbol{P}}^{(k)}f^{(k)}(\boldsymbol{X};\boldsymbol{\theta}^*) - \bar{\boldsymbol{y}}\right\|_2^2 &\leq \left\|\bar{\boldsymbol{P}}^{(k)}f(\boldsymbol{X};\boldsymbol{\theta}^*) - \bar{\boldsymbol{y}}\right\|_2^2 \\
&= \left\|\left(\bar{\boldsymbol{P}}^{(k)} - \bar{\boldsymbol{P}}^*\right)f(\boldsymbol{X};\boldsymbol{\theta}^*) + \bar{\boldsymbol{P}}^*f(\boldsymbol{X};\boldsymbol{\theta}^*) - \bar{\boldsymbol{y}}\right\|_2^2 \\
&\leq \left\|\left(\bar{\boldsymbol{P}}^{(k)} - \bar{\boldsymbol{P}}^*\right)f(\boldsymbol{X};\boldsymbol{\theta}^*)\right\|_2^2 + \left\|\bar{\boldsymbol{P}}^*f(\boldsymbol{X};\boldsymbol{\theta}^*) - \bar{\boldsymbol{y}}\right\|_2^2 \\
&\quad + 2\left\|\left(\bar{\boldsymbol{P}}^{(k)} - \bar{\boldsymbol{P}}^*\right)f(\boldsymbol{X};\boldsymbol{\theta}^*)\right\|_2\left\|\bar{\boldsymbol{P}}^*f(\boldsymbol{X};\boldsymbol{\theta}^*) - \bar{\boldsymbol{y}}\right\|_2.
\end{aligned} \quad (28)$$

Adding (27) and (28) then leads to

$$\begin{aligned}
\left\|\bar{\boldsymbol{P}}^{(k)}f\left(\boldsymbol{X};\boldsymbol{\theta}^{(t)}\right) - \bar{\boldsymbol{y}}\right\|_2^2 &\leq \frac{\sigma^{(k)}}{2\phi(t+1)^2}\left\|f(\boldsymbol{X};\boldsymbol{\theta}^*)\right\|_2^2 + \left\|\left(\bar{\boldsymbol{P}}^{(k)} - \bar{\boldsymbol{P}}^*\right)f(\boldsymbol{X};\boldsymbol{\theta}^*)\right\|_2^2 \\
&\quad + \left\|\bar{\boldsymbol{P}}^*f(\boldsymbol{X};\boldsymbol{\theta}^*) - \bar{\boldsymbol{y}}\right\|_2^2 + 2\left\|\left(\bar{\boldsymbol{P}}^{(k)} - \bar{\boldsymbol{P}}^*\right)f(\boldsymbol{X};\boldsymbol{\theta}^*)\right\|_2\left\|\bar{\boldsymbol{P}}^*f(\boldsymbol{X};\boldsymbol{\theta}^*) - \bar{\boldsymbol{y}}\right\|_2.
\end{aligned} \quad (29)$$

And by changing the order of terms, we can produce

$$
\begin{aligned}
\left\| \bar{P}^{(k)} f\left(X;\theta^{(t)}\right) - \bar{y} \right\| &- \left\| \bar{P}^* f\left(X;\theta^*\right) - \bar{y} \right\|_2^2 \\
&\leq \frac{\sigma^{(k)}}{2\phi(t+1)^2} \left\| f(X;\theta^*) \right\|_2^2 + \left\| \left(\bar{P}^{(k)} - \bar{P}^*\right) f\left(X;\theta^*\right) \right\|_2^2 \\
&\quad + 2\left\| \left(\bar{P}^{(k)} - \bar{P}^*\right) f\left(X;\theta^*\right) \right\|_2 \left\| \bar{P}^* f\left(X;\theta^*\right) - \bar{y} \right\|_2 \\
&\leq \frac{\sigma^{(k)}}{2\phi(t+1)^2} \left\| f(X;\theta^*) \right\|_2^2 + \left\| \left(\bar{P}^{(k)} - \bar{P}^*\right) \right\|_2^2 \left\| f\left(X;\theta^*\right) \right\|_2^2 \\
&\quad + 2\left\| \left(\bar{P}^{(k)} - \bar{P}^*\right) \right\|_2 \left\| f\left(X;\theta^*\right) \right\|_2 \left\| \bar{P}^* f\left(X;\theta^*\right) - \bar{y} \right\|_2 \\
&\leq O\left(\frac{1}{t^2}\right) + O\left(e^{-\frac{\mu}{\sigma}k}\right) + O\left(e^{-\frac{\mu}{2\sigma}k}\right) \\
&\leq O\left(\frac{1}{t^2} + e^{-ck}\right),
\end{aligned} \tag{30}
$$

where $c = \frac{\mu}{\sigma}$ and we have applied (25) and (26) in producing the second-to-last inequality.

And finally, since $f\left(X;\theta^*\right)$ is the optimal solution of $\ell_\theta^*$, we also have

$$
\begin{aligned}
\left\| \bar{P}^* f\left(X;\theta^*\right) - \bar{y} \right\|_2^2 &\leq \left\| \bar{P}^* f\left(X;\theta^{(t)}\right) - \bar{y} \right\|_2^2 \\
&= \left\| \left(\bar{P}^* - \bar{P}^{(k)}\right) f\left(X;\theta^{(t)}\right) \right\|_2^2 + \left\| \bar{P}^{(k)} f\left(X;\theta^{(t)}\right) - \bar{y} \right\|_2^2 \\
&\quad + 2\left\| \left(\bar{P}^* - \bar{P}^{(k)}\right) f\left(X;\theta^{(t)}\right) \right\|_2 \left\| \bar{P}^{(k)} f\left(X;\theta^{(t)}\right) - \bar{y} \right\|_2.
\end{aligned} \tag{31}
$$

And by changing the order of terms, we arrive at

$$
\begin{aligned}
\left\| \bar{P}^{(k)} f\left(X;\theta^{(t)}\right) - \bar{y} \right\|_2^2 &- \left\| \bar{P}^* f\left(X;\theta^*\right) - \bar{y} \right\|_2^2 \\
\geq &-\left\| \left(\bar{P}^* - \bar{P}^{(k)}\right) f\left(X;\theta^{(t)}\right) \right\|_2^2 - 2\left\| \left(\bar{P}^* - \bar{P}^{(k)}\right) f\left(X;\theta^{(t)}\right) \right\|_2 \left\| \bar{P}^{(k)} f\left(X;\theta^{(t)}\right) - \bar{y} \right\|_2 \\
\geq &-O(e^{-ck}).
\end{aligned} \tag{32}
$$

Combining (30) and (32), we have

$$
-O(e^{-ck}) \leq \ell_\theta^{(k)}(\theta^{(t)}) - \ell_\theta^*(\theta^*) \leq O\left(\frac{1}{t^2} + e^{-ck}\right),
$$

from which it follows that

$$
\left| \ell_\theta^{(k)}(\theta^{(t)}) - \ell_\theta^*(\theta^*) \right| \leq O\left(\frac{1}{t^2} + e^{-ck}\right) \tag{33}
$$

which concludes the proof. $\qquad\square$

To prove the Corollary 1, we introduce Lemma 3 firstly.

**Lemma 3.** *Let $\mathcal{D}(u) = \|u\|_2^2$, $g(u) = u$, $\alpha = \|\lambda L + I\|_2^{-1}$ in the loss from (7). Additionally, a penalty factor $\varepsilon\|f(X;\theta)\|_{\mathcal{F}}^2$ is added to (7), then we have $\ell_\theta^{(k)}$ is $\sigma^{(k)}$-smooth and $\mu$-strongly convex, where $\sigma^{(k)} = \sigma_{\max}(P_{1:m}^{(k)})$ and $\mu = \epsilon$. Moreover, we have $0 < \mu < \sigma^{(k)} < +\infty$ for any $k \geq 0$.*

The proof is also straightforward given the stated assumptions, the loss from (7) reduces to $\ell_\theta^{(k)}(\boldsymbol{\theta}) =$

$$\sum_{i=1}^{m}\left(\boldsymbol{P}^{(k)}\left[f\left(\boldsymbol{X};\boldsymbol{\theta}\right)\right]_i - \boldsymbol{y}_i\right)^2 + \varepsilon\|f(\boldsymbol{X};\boldsymbol{\theta})\|_{\mathcal{F}}^2 = \left\|\begin{bmatrix}\boldsymbol{P}_{1:m}^{(k)}\\\sqrt{\varepsilon}\boldsymbol{I}_n\end{bmatrix}f\left(\boldsymbol{X};\boldsymbol{\theta}\right) - \begin{bmatrix}\boldsymbol{y}_{1:m}\\\boldsymbol{0}\end{bmatrix}\right\|_2^2. \quad (34)$$

Denote $\widetilde{\boldsymbol{P}}^{(k)} = \begin{bmatrix}\boldsymbol{P}_{1:m}^{(k)}\\\sqrt{\varepsilon}\boldsymbol{I}_n\end{bmatrix}$, then we have $\ell_\theta^{(k)}$ is $\sigma^{(k)}$-smooth and $\mu$-strongly convex, where $\sigma^{(k)} = \sigma_{\max}(\widetilde{\boldsymbol{P}}^{(k)})$ and $\mu = \epsilon$.

The proof of Corollary 1 resembles the proof of Theorem 1. Since the bilevel loss (7) is $\sigma$-smooth and $\mu$-strongly convex, we can apply gradient boosting machine convergences results from Lu & Mazumder (2020)[Theorem 4.1] : let $\ell_\theta^{(k)}$ be $\sigma^{(k)}$-smooth and $\mu$-strongly convex, consider Gradient Boosting Machine with constant step-size rule with $\eta = 1/\sigma^{(k)}$. If $\Gamma$ denotes the value of the corresponding MCA, then for all $t \geq 0$ we have that

$$\ell_\theta^{(k)}(\boldsymbol{\theta}^{(t)}) - \ell_\theta^{(k)}(\boldsymbol{\theta}^*) \leq (1 - \frac{\mu}{\sigma^{(k)}}\Gamma^2)^t\left(\ell_\theta^{(k)}(\boldsymbol{\theta}^{(0)}) - \ell_\theta^{(k)}(\boldsymbol{\theta}^*)\right). \quad (35)$$

The remaining part of proof is the same with the proof of Theorem 1.

## ADDITIONAL REFERENCES FOR THE SUPPLEMENTARY

S. Bubeck. Convex optimization: Algorithms and complexity. *arXiv preprint arXiv:1405.4980*, 2014.

S. Ivanov and L. Prokhorenkova. Boost then convolve: Gradient boosting meets graph neural networks. *arXiv preprint arXiv:2101.08543*, 2021.

H. Lu and R. Mazumder. Randomized gradient boosting machine. *SIAM Journal on Optimization*, 30(4):2780–2808, 2020.

H. Lu, S. P. Karimireddy, N. Ponomareva, and V. Mirrokni. Accelerating gradient boosting machines. In *International Conference on Artificial Intelligence and Statistics*, pages 516–526. PMLR, 2020.

M. Wang, D. Zheng, Z. Ye, Q. Gan, M. Li, X. Song, J. Zhou, C. Ma, L. Yu, Y. Gai, et al. Deep graph library: A graph-centric, highly-performant package for graph neural networks. *arXiv preprint arXiv:1909.01315*, 2019.

