# OpenReview forum: "Does your graph need a confidence boost?  Convergent boosted smoothing on graphs with tabular node features"
_ICLR.cc/2022/Conference — ICLR 2022 Spotlight_

### Official Review · Reviewer_zcK6 · 2021-10-31

**Correctness:** 4
**Technical Novelty And Significance:** 2
**Empirical Novelty And Significance:** 3
**Recommendation:** 6
**Confidence:** 4

**Main Review:**

Strength
- The approach nicely defines a single objective that the model (graph propagation + decision trees) optimizes.
- Empirically, there is a nice improvement over the existing BGNN.

Weakness:
- The studied problem does not seem particularly novel to me, especially given BGNN. Given BGNN, the scope seems a bit narrow to me (although I acknowledge that the authors solve the problem in a potentially better way than the BGNN paper).

Comments/questions:
- I am curious to see the result of XGBoost + C&S (e.g., use XGBoost as the base predictor in C&S).
- Does the framework supports any propagation rules beyond (6)? I would be curious to see how general the method is.


**Summary Of The Paper:**

In this paper, the authors present a new approach to combine the boosted decision tree classifiers with a graph propagation model, which is important in handling table input data. The approach casts the graph propagation as an optimization problem, where the input node features are generated by boosted decision trees. The gradient can be taken in the functional space to learn the decision trees to minimize a unified loss. The final algorithm is shown to minimize the unified loss in a principled manner. The superior performance is demonstrated over the existing BGNN model.


**Summary Of The Review:**

Overall, the approach seems sound and principled, although the scope is a bit narrow. Hence, I will give the weak accept. I would also like the authors to address my comments/questions.

---

> ### Author Response · Authors · 2021-11-18
> **Response to Reviewer zcK6**
>
> Thanks for the detailed comments. We address each one below.
>
> **Questions**: The studied problem does not seem particularly novel to me, especially given BGNN. Given BGNN, the scope seems a bit narrow to me (although I acknowledge that the authors solve the problem in a potentially better way than the BGNN paper).
>
> **Response**:  We certainly agree that the BGNN paper deserves the credit for first addressing the problem of combining boosting with GNNs for handling tabular data.  That being said, we still believe that there remain important directions for innovation in this space (especially given the vast relevance of tabular data with attendant relations across numerous industry applications).  For example, our specific design of an integrated bi-level loss that facilitates convergence guarantees and simplified yet performant practical deployment can be viewed as novel contributions.
>
>
> **Questions**: I am curious to see the result of XGBoost + C&S (e.g., use XGBoost as the base predictor in C&S).
>
> **Response**: This is a good suggestion.  Actually we have previously tried C&S with various forms of boosting as the base predictor.  The results are very similar to the CatBoost+ baseline reported in our paper.
>
>
> **Questions**: Does the framework supports any propagation rules beyond (6)? I would be curious to see how general the method is.
>
>
> **Response**: Yes, our framework supports a broad family of general propagation rules. Please see Section 3.4 of our submission which presents several possibilities.  Even so, the simple rule from equation (6) performs well without adding extra hyperparameters.

---

> > ### Comment · Reviewer_zcK6 · 2021-11-18
> > **Suggestion to add XGBoost + C&S results**
> >
> > Thank you for the response! Could you please actually include the results of XGBoost + C&S? I believe this would help the paper, as people think C&S + strong base predictor is the state-of-the-art. For C&S, there are two hyper-smoothing parameters you need to carefully tune in order to achieve the best performance. Thanks.

---

> > > ### Author Response · Authors · 2021-11-19
> > > **Response to Reviewer zcK6**
> > >
> > > **Question**: Could you please actually include the results of XGBoost + C&S? I believe this would help the paper, as people think C&S + strong base predictor is the state-of-the-art. For C&S, there are two hyper-smoothing parameters you need to carefully tune in order to achieve the best performance.
> > >
> > > **Response**: Thanks for the response and question. Per the reviewer's suggestion, we have quickly run XGBoost + C&S, with all hyperparameters for XGBoost and C&S carefully tuned to achieve the best performance on the node regression benchmarks.  The trial-averaged RMSE results, compared with CatBoost+ and EBBS as reported in our paper, are as follows:
> > >
> > > |              | House | County | VK   | Avazu  |
> > > |--------------|-------|--------|------|--------|
> > > | XGBoost+C&S  | 0.55  | 1.25   | 6.97 | 0.1097 |
> > > | Catboost+    | 0.54  | 1.25   | 6.96 | 0.1083 |
> > > | EBBS         | 0.45  | 1.11   | 6.90 | 0.1062 |
> > >
> > > From these results we observe that XGBoost+C&S performance is nearly the same as CatBoost+ as expected given their analogous basic structure, and our end-to-end EBBS approach remains superior.  Note that prior to our submission we tried CatBoost and LightGBM combined with several different forms of graph propagation post-processing, including C&S.  And generally speaking the results were similar.  That being said, C&S is definitely a powerful method, and as new tabular benchmarks become available in the future it may offer a more pronounced advantage (one that EBBS can actually exploit as well via convergent end-to-end training; this involves combining  results from Sections 3.1 and 3.4 of our submission). In any event, we had not previously experimented with XGBoost, so XGBoost+C&S was nonetheless a useful benchmark to test; good suggestion.

---

> > > > ### Author Response · Authors · 2021-11-19
> > > > **One quick follow-up point to Reviewer zcK6**
> > > >
> > > > Additionally, one quick follow-up point that may be worth addressing here.  The reviewer mentioned that a strong base model plus C&S can often achieve SOTA results.  In fact, our results are basically consistent with this assertion.  For example, the XGBoost base model plus C&S (see results in the added rebuttal table above) mostly outperforms all of the GNN baselines reported in Table 1.  And it is really only EBBS that consistently outperforms XGBoost+C&S via convergent, end-to-end training of a strong tabular+graph model.

---

### Official Review · Reviewer_omBE · 2021-10-31

**Correctness:** 3
**Technical Novelty And Significance:** 3
**Empirical Novelty And Significance:** 2
**Recommendation:** 6
**Confidence:** 2

**Main Review:**

Strengths:  This paper proposes EBBS, efficient bilevel boosted smoothing, a novel way to combine GNN and Gradient boosting for learning tabular graph data. The addressed problem of integrating boosting into GNNs is very interesting to me. Also learning over tabular graph data should receive a wide audience given its importance in the industry. Empirical experiments show EBBS outperforms baseline methods on multiple node classification and node regression datasets.

Weakness: In my opinion, the main weaknesses are in writing/presentation and reproducibility.
First, I feel the writing of Section 3 can be improved to avoid readers' confusion. For example
  - We can be more clear about how Eq 2 is rooted in Zhou et al 2004. In fact, I didn't get it when I checked the referenced paper.
  - In (2), are both Z and \theta learnable?
  - P is binded twice, once in P* (Eq 3) and once in P^{k} (Eq 6)
  - In Eq 7, what is for inner level optimization and what is for outer level optimization?
  - Is "Graph-Aware Propagation Layers" terminology used in the literature?
Second, it seems that the proposed method EBBS will be incorporating test nodes during training. Will this cause test information to leak into the training process? Is there any specific preprocessing to avoid leaking? Is EBBS easy to implement?


Minor issues (typos, formats):
- "on top model leaderboards" -> on the top of model leaderboards
- the template seems a bit different from the normal one, especially the font and the colour of the citation text
- "whereby the end-to-end training of a bilevel loss is such that values of the boosted base model f ebb and flow across the input graph producing a smoothed predictor f" -> not sure if there is a grammar issue
- "use mi and to reference" -> delete "and"

**Summary Of The Paper:**

This paper proposes a new way to integrate graph-based models with boosting based on principled meta loss, named EBBS. In experiments, the proposed method outperforms tabular baselines, GNN baselines, and some hybrid strategies like BGNN over some node classification/regression datasets.

**Summary Of The Review:**

Overall, I like the problem the paper aims to address - how to better combine GNN with boosting methods for learning on tabular data. The paper proposes a novel way to address this problem, which is based on a principled meta-loss. Empirical results show the effectiveness of the results. I feel the paper can be improved more by iterating on the formulations in Sec 3.

---

> ### Author Response · Authors · 2021-11-18
> **Response to Reviewer omBE**
>
> Thanks for the detailed comments. We address each one below.
>
> **Questions**: We can be more clear about how Eq 2 is rooted in Zhou et al 2004. In fact, I didn't get it when I checked the referenced paper.
>
> **Response**: The key identity is $\mbox{tr}\left[Z^\top L Z \right] =  \sum_{\{i,j\} \in \mathcal{E} }\left\| z_i - z_j \right\|_2^2$, which is also discussed in the sentence beginning with "Intuitively, solutions of ..." in our submission.  The only relevant difference then is just that in Zhou et al. (2004) (see their equation 4) a weighted graph and a normalized Laplacian are used (the latter leads to the inverse square-root of the node degree factors), common substitutions that can also be seemlessly integrated within our EBBS framework if needed.
>
>
> **Questions**: In (2), are both Z and \theta learnable?
>
> **Response**: In equation (2), the only optimization variable is $Z$.  However, the optimal solution of (2) will be a function of $\theta$ as shown in equation (3).  And once we plug this optimal, $\theta$-dependent value into the meta-loss function from equation (7), we can then optimize over $\theta$.  This is the crux of our bilevel optimization scheme.  Another way to view this process is that the solution of (2) produces a set of features that are useful for making predictions by solving (3), and bilevel optimization allows for full end-to-end optimization of both features and the final predictor.
>
>
> **Questions**: P is binded twice, once in P* (Eq 3) and once in P^{k} (Eq 6).
>
> **Response**:  We may have misunderstood the question, but equation (3) provides a specific definition for $P^*$, while in equation (6) the notation $P^{(k)}$ is used to denote an approximation to $P^*$ obtained by conducting $k$ gradient steps (which can be computed via equation (5); please also see text below equation (6)).
>
>
> **Questions**: In Eq 7, what is for inner level optimization and what is for outer level optimization?
>
> **Response**: In equation (7), $\widetilde{f}^{(k)}\left(X; \theta \right)$ denotes a graph regularized version of the base estimator $f^{(k)}\left(X; \theta \right)$, and it is the approximated minimum solution of energy equation (2). This can be referred to as the inner-level optimization. In contrast, the outer-level optimization is an application-specific meta-loss minimization problem, e.g regression or classification task. We specify the meta-loss function in equation (7).
>
>
> **Question**: Is "Graph-Aware Propagation Layers" terminology used in the literature?
>
> **Response**: This is a descriptive phrase as opposed to formal terminology, although similar phrasings can be found in the literature.  Perhaps more completely though, a graph-aware propagation layer simply refers to a graph-dependent function that smooths (i.e., propagates) an input vector across the edges of a graph.
>
> **Question**: it seems that the proposed method EBBS will be incorporating test nodes during training. Will this cause test information to leak into the training process? Is there any specific preprocessing to avoid leaking?
>
> **Response**: Actually test node *labels* are never used during training; critically, only test node *input features* are utilized. This is because the test node input features contribute to the prediction function for nearby training node labels.  Consequently, there is no data leakage issue, and the function-space gradient signal used for training is completely independent of test node labels.
>
>
> **Question**: Is EBBS easy to implement?
>
> **Response**: EBBS is easy to implement, which is one of its attractive qualities.  The code will be released after final decisions.

---

> > ### Comment · Reviewer_omBE · 2021-11-29
> > **Thank you for the response!**
> >
> > Thank you for your response! I have read the response and would like to remain in support of this paper.

---

### Official Review · Reviewer_ahft · 2021-11-03

**Correctness:** 4
**Technical Novelty And Significance:** 3
**Empirical Novelty And Significance:** 3
**Recommendation:** 8
**Confidence:** 3

**Details Of Ethics Concerns:**

The paper does not introduce any new ethics concerns.

**Main Review:**

**Summary** This paper investigates tabular, graph-based data for classification and regression tasks. The proposed approach is an end-to-end, bilevel, combination of label propagation and boosting. The authors contribute not only an empirical analysis of the proposed approach on 8 datasets demonstrating the effectiveness of the proposed approach as well as a theoretical analysis.

**Merits** I believe that this is a strong paper that clearly outlines a proposed approach for boosting in this non-iid setting of graph data. The proposed approach has a convergence guarantee and is shown to be very effective empirically. Overall, this seems to be a strong result. The supplemental material seems to give statistical significance of the improvements shown. Compared to the best previous method BGNN, combining boosting and GNNs, EBBS achieves stronger empirical results. The algorithms and theoretical results are discussed as well.

**Weaknesses** Here are a few concerns / suggestions:
* It could perhaps be made stronger by including some of the additional analysis that is in the supplemental material that investigates the trade-offs and ablations of the approaches, in the main body of the text.
* I think that the paper could be made much stronger with a simple motivating (perhaps synthetic) example that illustrates where and when EBBS can be useful compared to competing methods. While convergence guarantees and motivations are described, a clear simple example (which might further be useful in using ablations to identify contributions of different parts of the solution) could strength the paper.

**Minor Notes**
* Why is "GraphData" one word in the title?
* Figure 1 would be easier to read if Y-axis was the same in both plots

**Summary Of The Paper:**

Following in the success of boosting methods for tabular data, this paper introduces a new boosting approach for data that graph-based with tabular features. The proposed approach, efficient bilevel boosted smoothing (EBBS), has convergence guarantees as well as empirical successes compared to competing methods.

**Summary Of The Review:**

This paper provides both theoretical and empirical results for a boosting method for graph structured data. The results appear to advance the state of the art and the submission seems to have valuable contributions.

---

> ### Author Response · Authors · 2021-11-18
> **Response to Reviewer ahft**
>
> Thanks for the detailed comments. We address each one below.
>
> **Questions**: It [the paper] could perhaps be made stronger by including some of the additional analysis that is in the supplemental material that investigates the trade-offs and ablations of the approaches, in the main body of the text.
>
> **Response**: We agree with this suggestion; however, unfortunately the main paper already consumes the full 9 pages and this year ICLR does not grant additional space for addressing reviewer recommendations.
>
>
> **Questions**: I think that the paper could be made much stronger with a simple motivating (perhaps synthetic) example that illustrates where and when EBBS can be useful compared to competing methods. While convergence guarantees and motivations are described, a clear simple example (which might further be useful in using ablations to identify contributions of different parts of the solution) could strength the paper.
>
> **Response**: Perhaps one motivational example is the additional stability loosely afforded by our convergent algorithm when applied under non-ideal testing conditions.  For example, in Figure 2 we observe how the training curve of EBBS remains stable even though the hyperparameters were simply borrowed from a different dataset.  Beyond this though, we unfortunately don't have space to introduce and analyze a new synthetic model.

---

### Official Review · Reviewer_Mnik · 2021-11-03

**Correctness:** 3
**Technical Novelty And Significance:** 4
**Empirical Novelty And Significance:** 3
**Recommendation:** 8
**Confidence:** 3

**Main Review:**

The notation used is easy to understand, as is the mathematical explanation in section 3, which is presented in a comprehensive but concise manner.
"for EBBS we must fit the weak-learners to gradients from both the training and test nodes". This is the sentence that I am most concerned about, as the use of test data in the training phase may render the results obtained invalid. Although the authors give their own explanation of why the test nodes should also be used during training, i.e. for the propagation of information in the graph, if the test labels are used during the train there is no longer any separation between train and test. I have this doubt because the labels are used to calculate function-space gradients. Is this correct?

Algorithm 1 could be described in a little more detail.

The analysis of the convergence of the method by theorem is very good. The remarks connected to the theorem are interesting, but could have been treated in more detail (they are in part in the supplements).
If experiments are done with different random seeds (as stated), then results in Table 1 should be reported with their corresponding standard deviation or confidence interval.
Why are some results reported with different decimal places in the table? I am talking specifically about the CS column, but also for the Slap, DBLP and Phy columns one decimal place could be added.
If the datasets were taken from Ivanov & Prokhorenkova (2021), why was Wiki not taken? The reason should be that, being a homogeneous dataset, then, as explained by Ivanov & Prokhorenkova (2021), "neural network approaches are sufficient to achieve the best results". It would still be interesting as a comparison. Also as a comparison with them, the House and VK datasets could also be used for classification. They also report the standard deviation of all results.
Also, the results in the table match those of Ivanov & Prokhorenkova (2021), but I do not understand why their LightGBM results row has become the CatBoost row in this article for the Slap, DBLP and OGB-ArXiv datasets. Is this perhaps an error?
Although the difference between OGB-ArXiv and the other datasets is properly explained, I think it still makes sense to put those results together with the others in Table 1.
"Although tabular graph data for node classification is widely-available in industry, unfortunately there is currently little publicly-available, real-world data that can be used for benchmarking."
This sentence is very vague and I am not fully convinced of its veracity.

The mention of the method called CatBoost+ is interesting, but it is given too little space. Why is it not considered in "ours"? If the idea is picked up by some other work, let it be mentioned properly.
"revealing that it may be more robust to non-ideal use cases". That's why it might be interesting to add homogenous datasets and see if it applies there too.
While in the main part it says "This suggests that in new application domains it may conceivably be easier to adapt EBBS models", in the supplements it says "It shows EBBS can be run with mostly shared hyperparameters across all datasets". I don't think there are enough experiments/results to say that, but I'd stick with "suggest" in the supplementary materials as well. Maybe add a sentence about the possibility of exploring this area more in future work.

After the rebuttal I have strengthen my opinion on the quality of the paper. I believe it is a nice contribution to the field.

**Summary Of The Paper:**

The authors propose a new method for integrating graph-based models with boosting. This is done using the typical method involving residuals and weak-learners, but adding a step where information is propagated in the graph. The approach is also simple, as no GNNs or other auxiliary models are required. It is also shown how the meta-loss introduced by the authors provides convergence given some moderate assumptions. According to the experiments reported, the proposed model is better than the current state of the art in the considered domain.

**Summary Of The Review:**

The work is well structured, with a good theoretical basis to support the proposed methodology. The empirical results are very promising, although the small amount of datasets combined with the lack of confidence intervals does not allow for meaningful conclusions to be drawn.
The only major doubt concerns the use of test data in the training phase, which may have compromised the whole experiment.

---

> ### Author Response · Authors · 2021-11-18
> **Response to Reviewer Mnik Part 1**
>
> Thanks for the detailed comments. We address each one below.
>
> **Questions**: The notation used is easy to understand, as is the mathematical explanation in section 3, which is presented in a comprehensive but concise manner. "for EBBS we must fit the weak-learners to gradients from both the training and test nodes". This is the sentence that I am most concerned about, as the use of test data in the training phase may render the results obtained invalid. Although the authors give their own explanation of why the test nodes should also be used during training, i.e. for the propagation of information in the graph, if the test labels are used during the train there is no longer any separation between train and test. I have this doubt because the labels are used to calculate function-space gradients. Is this correct?
>
> **Response**: Actually test node *labels* are never used during training; critically, only test node *input features* are utilized. This is because the test node input features contribute to the prediction function for nearby training node labels.  Consequently, there is no data leakage issue, and the function-space gradient signal used for training is completely independent of test node labels.
>
> **Questions**: Algorithm 1 could be described in a little more detail.  The analysis of the convergence of the method by theorem is very good. The remarks connected to the theorem are interesting, but could have been treated in more detail (they are in part in the supplements).
>
> **Response**: This year it appears that the page length for ICLR will not be extended to allow for extra content addressing reviewer concerns.  So while we agree with the reviewer that more detail in these areas would be quite helpful, unfortunately we do not presently have sufficient space outside of the supplementary.
>
>
> **Questions**: If experiments are done with different random seeds (as stated), then results in Table 1 should be reported with their corresponding standard deviation or confidence interval. Why are some results reported with different decimal places in the table? I am talking specifically about the CS column, but also for the Slap, DBLP and Phy columns one decimal place could be added.
>
>
> **Response**: Comprehensive discussion of standard errors and statistical significance was originally deferred to the supplementary to save space, noting that the Table 1 caption provided this reference.  However, we agree with the reviewer that it could be nice to include at least some of these results in the main paper, which only strengthens the support for EBBS.  Therefore, while it is hard to expand Table 1 to include all details within the page limit, we have added a new small Figure 1 and attendant discussion to demonstrate how our approach actually achieves the best performance across all random trials.
>
>
> Regarding the decimal place issue, this was our formatting mistake, and we have made them all consistent in the revision.  Thanks for catching this.
>
>
> **Questions**: If the datasets were taken from Ivanov & Prokhorenkova (2021), why was Wiki not taken? The reason should be that, being a homogeneous dataset, then, as explained by Ivanov & Prokhorenkova (2021), "neural network approaches are sufficient to achieve the best results". It would still be interesting as a comparison.
>
> **Response**: We did not run comparisons on the Wiki data in the original submission because it is a regression task with homogeneous node features, and therefore we did not feel it was an important benchmark for tabular+graph models (we did however include OGB-ArXiv results in Table 2, largely because this is a very famous node classification benchmark). Indeed, from reference Ivanov & Prokhorenkova (2021) Table 2, the BGNN model was not competitive on the Wiki data, being outperformed by most GNN models. Regardless, per the reviewer's comment, we have now tried running our EBBS model on the Wiki data with splits from Ivanov & Prokhorenkova (2021), and the trial-averaged performance is 41432 RMSE, which is better than BGNN's 48119 RMSE.  Note that the average EBBS improvement is 6687, which is significant given that the stdev of the gap across trials is only 3473 (and EBBS has lower error across all trials).

---

> > ### Author Response · Authors · 2021-11-18
> > **Response to Reviewer Mnik Part 2**
> >
> > **Questions**: Also as a comparison with them, the House and VK datasets could also be used for classification. They also report the standard deviation of all results.
> >
> >
> > **Response**: House_class and VK_class are actually redundant, synthetic classification datasets. More specifically, in the BGNN paper Ivanov & Prokhorenkova (2021), the original House and VK regression datasets were merely modified by converting numerical target labels into several discrete classes by thresholding. Thus we omitted these two artificial datasets and instead employed two real classification datasets popular in the GNN literature: Coauthor-CS and Coauthor-Phy.
> >
> >
> > **Questions**: Also, the results in the table match those of Ivanov & Prokhorenkova (2021), but I do not understand why their LightGBM results row has become the CatBoost row in this article for the Slap, DBLP and OGB-ArXiv datasets. Is this perhaps an error?
> >
> > **Response**: Yes, this discrepancy seems to originate from the original arxiv version; please refer to the camera-ready version of Ivanov & Prokhorenkova (2021).
> >
> >
> >
> > **Questions**: "Although tabular graph data for node classification is widely-available in industry, unfortunately there is currently little publicly-available, real-world data that can be used for benchmarking." This sentence is very vague and I am not fully convinced of its veracity.
> >
> > **Response**: As future work, we are looking to create more suitable classification benchmarks that combine tabular node features with graphs. This is critical given that, while industry is awash in such datasets (or the means of extracting such datasets from widely-prevalent  relational databases), representative publicly-available benchmarks are slim to none.
> >
> >
> > **Questions**: The mention of the method called CatBoost+ is interesting, but it is given too little space. Why is it not considered in "ours"? If the idea is picked up by some other work, let it be mentioned properly. "revealing that it may be more robust to non-ideal use cases". That's why it might be interesting to add homogenous datasets and see if it applies there too. While in the main part it says "This suggests that in new application domains it may conceivably be easier to adapt EBBS models", in the supplements it says "It shows EBBS can be run with mostly shared hyperparameters across all datasets". I don't think there are enough experiments/results to say that, but I'd stick with "suggest" in the supplementary materials as well. Maybe add a sentence about the possibility of exploring this area more in future work.
> >
> >
> >
> > **Response**: Yes, CatBoost+ can be considered as our method, at least in the sense that prior work has not proposed this specific baseline. Actually, EBBS can be viewed as a way of training CatBoost+ in an end-to-end fashion. Given that our paper's main contribution is to propose EBBS and provide related convergence analysis, we did not talk too much about Catboost+ to accommodate the limited space.
> >
> > Additionally, in terms of further testing with homogeneous node features, please see our response from above that mentions new results with Wiki data.  We have also edited the wording in the supplementary per the reviewers suggestion, and reiterate that both Figure 2 (main paper) and Figure S1 (supplementary) support the notion that it may be easier to adapt EBBS models for new use cases.

---

### Decision · Program_Chairs · 2022-01-20

**Decision:**

Accept (Spotlight)

**Comment:**

The paper addresses a problem encountered in many real-world applications, i.e. the treatment of tabular data, composed of heterogeneous feature types, where samples are not i.i.d. In this case, learning is more effective if the typically successful approach for i.i.d. data (boosted decision trees + committee techniques) is combined with GNN to take into account the dependencies between samples. The main contribution of the paper with respect to previous work in the field is the introduction of a principled approach to pursue such integration. One important component of the proposed approach is played by the definition of a specific bi-level loss (efficient bilevel boosted smoothing) that allows for convergence guarantees under mild assumptions. Both theoretical and experimental contributions are sound and convincing, justifying the claimed merits of the proposed approach. Another strong point is the fact that the proposed approach is general and amenable to support a broad family of propagation rules. One weakness with the original submission was presentation, mainly because some key information was confined into the supplementary material. The revised version addressed this problem and added some more empirical results that confirmed the superiority of the proposed approach.
Finally, the fact that learning over tabular graph data is very important in Industry, the proposed approach may be of interest for a wide audience.